# Association of serum lysophosphatidylcholine acyltransferase 3 levels with metabolic variables and risk of type 2 diabetes mellitus: A cross-sectional study

Haifeng Zhu[1☯], Ziyi Zhong[2☯], Jing Jin[1☯], Wei Liu[1], Yuan Cao[1], Yawen Guo[1], Gaonian Zhao[ID][1‡]*, Qian Li[2‡]*

1 Rehabilitation Medicine Department, Taizhou People's Hospital, Taizhou City, Jiangsu Province, China,
2 Endocrinology Department, Nanjing First Hospital, Nanjing Medical University, Nanjing City, Jiangsu Province, China

☯ These authors contributed equally to this work.
‡ GZ and QL also contributed equally to this work.
* tzsrmyy01@163.com (GZ); njdyyynfm@163.com (QL)

## Abstract

### Aims

This study aimed to explore the role of serum lysophosphatidylcholine acyltransferase 3 (LPCAT3) in glucose and lipid metabolism and its association with type 2 diabetes mellitus (T2DM).

### Methods

Between July and December 2024, we recruited 256 newly diagnosed T2DM patients and 252 gender- and age-matched individuals with normal glucose tolerance (NGT). Serum LPCAT3 levels were measured using ELISA. Group comparisons were conducted via t-tests or Mann-Whitney U tests. Spearman correlation analysis assessed the relationship between LPCAT3 and metabolic variables. Linear regression identified independent predictors of LPCAT3 levels. Partial Least Squares (PLS) analysis evaluated the correlations between serum LPCAT3 and obesity-related anthropometric indicators, blood glucose and lipid indicators. Logistic regression evaluated the association between LPCAT3 levels and T2DM risk, and ROC analysis determined its predictive value.

### Results

Median LPCAT3 level was lower in T2DM patients (21.51 ng/ml, IQR: 8.47–35.63) compared to the NGT group (24.43 ng/ml, IQR: 14.41–49.37). In NGT individuals, LPCAT3 negatively correlated with high-density lipoprotein cholesterol (HDL), fasting blood glucose (FBG), and glycated hemoglobin (HbA1c). In T2DM patients, LPCAT3

**Data availability statement:** DOI: 10.17632/tjrdk8hmmb.2 (https://data.mendeley.com).

**Funding:** The author(s) received no specific funding for this work.

**Competing interests:** The authors have declared that no competing interests exist.

negatively correlated with body mass index (BMI) and waist circumference (WC). Linear regression identified BMI, HDL, and FBG as negative predictors of LPCAT3. PLS analysis revealed negative correlations between LPCAT3 and BMI, WC, HDL and FBG, but with large standard errors. When stratified by LPCAT3 tertiles, the lowest tertile initially showed a higher T2DM incidence than the highest tertile. However, after adjusting for obesity-related indicators, no significant difference was found between them. ROC analysis yielded an AUC of 0.580 for LPCAT3.

## Conclusion

Although serum LPCAT3 levels are lower in T2DM patients, its predictive capacity for T2DM is constrained. Moreover, the association between LPCAT3 and T2DM risk is likely confounded by obesity-related factors. While LPCAT3 tends to negatively correlate with BMI, HDL, and FBG, these correlations are complex and unstable.

## Introduction

Type 2 diabetes mellitus (T2DM) stands as one of the most widespread metabolic disorders worldwide, with its incidence steadily escalating, posing a formidable challenge to global public health [1]. Data from the International Diabetes Federation (IDF) reveals that over half a billion individuals globally are afflicted by diabetes, with T2DM accounting for approximately 90% of these cases [2]. The core pathophysiological characteristics of T2DM include insulin resistance and/or dysfunction of β-cells. These disruptions upset the delicate equilibrium crucial for blood glucose regulation, subsequently triggering a series of complications that significantly impact patients' quality of life and longevity [3].

With the progress in metabolomics and lipidomics research, the critical role of lipid metabolism dysregulation in the onset and progression of T2DM has gained increasing recognition [4,5]. Within the diverse lipid metabolic pathways associated with T2DM, the regulation of phospholipids, especially those involving lysophosphatidylcholines, has emerged as a focal point of interest [6]. Lysophosphatidylcholine acyltransferase 3 (LPCAT3), a pivotal enzyme in phospholipid biosynthesis, stands out as the most abundant member of the LPCAT family in the liver. It significantly affects membrane phospholipid composition, cellular signal transduction, and lipid metabolic pathways [7,8]. Recent investigations have further clarified that a deficiency of LPCAT3 in metabolic tissues, including the liver, skeletal muscle, and adipose tissue, can augment insulin signaling, enhance insulin sensitivity, and result in improved glucose tolerance [9–11].

However, most previous studies have primarily focused on the role of LPCAT3 within tissue-specific contexts, and there remains a significant gap in research regarding the function of serum LPCAT3. Given the potential importance of serum LPCAT3 in glucose and lipid metabolism, coupled with the escalating global burden of T2DM, it is worthwhile to explore the relationship between serum LPCAT3 levels and metabolic variables, as well as its potential predictive value for T2DM risk.

Delving into these associations may offer valuable insights into the pathogenesis of T2DM and potentially pave the way for the identification of novel biomarkers that may aid in early diagnosis and intervention.

In this study, the primary objective is to explore the association between serum LPCAT3 levels and the risk of T2DM. The secondary objectives include examining the correlations between serum LPCAT3 levels and various metabolic variables, such as blood glucose indicators (e.g., fasting blood glucose (FBG), glycated hemoglobin (HbA1c)), blood lipid indicators (e.g., high-density lipoprotein cholesterol (HDL)), and obesity-related anthropometric indicators (e.g., body mass index (BMI), waist circumference (WC)).

## Materials and methods

### Study subjects

This was a cross-sectional study conducted between July 1, 2024, and December 31, 2024, recruiting participants from four healthcare institutions: Taizhou People's Hospital, Taizhou Third People's Hospital, *Sixiang* People's Hospital, and *Ruici* Medical Center. The case group consisted of 256 patients newly diagnosed with T2DM, while the control group comprised 252 individuals with normal glucose tolerance (NGT), carefully matched for age and gender. The diagnosis of diabetes was established according to the 2024 criteria set by the American Diabetes Association (ADA) [12], which include a fasting blood glucose (FBG) level of ≥7.0 mmol/L, a 2-hour postprandial glucose (2hPG) level of ≥11.1 mmol/L during an oral glucose tolerance test (OGTT), or a glycated hemoglobin (HbA1c) level of ≥6.5%. Conversely, NGT was defined as having an FBG level of <5.6 mmol/L, a 2hPG level of <7.8 mmol/L, and an HbA1c level of <5.7%. The study included adults aged 18–80 years who had not used any medication in the preceding month and were classified into either a case group (newly diagnosed T2DM patients) or a control group (individuals with NGT). Participants were excluded if they met any of the following criteria: (1) other forms of diabetes (e.g., type 1, gestational, or secondary diabetes); (2) BMI ≥ 35 kg/m$^2$ or <18.5 kg/m$^2$; (3) smoking or alcohol consumption exceeding 140 g/week for males or 70 g/week for females; (4) pregnancy or lactation; (5) NYHA heart function class 2–4; (6) cirrhosis, transaminase levels >3 × the upper limit of normal; (7) estimated glomerular filtration rate (eGFR) <60 ml/(min·1.73m$^2$); (8) cancer; (9) active infections, or diabetic ketoacidosis; or (10) newly diagnosed diabetes patients with fasting C-peptide (CP) levels <200 pmol/L. The eGFR was calculated utilizing the Chronic Kidney Disease Epidemiology Collaboration (CKD-EPI) equation [13]. This study was approved by the Clinical Research Ethics Committee of Taizhou People's Hospital (approval number: KY-2024-072-01, approval date: June 13, 2024). The protocol was registered on the Chinese Clinical Trial Registry (registration number: ChiCTR2400086076, registration date: June 25, 2024). All participants provided written informed consent prior to enrollment. Initially, the consent did not cover ultrasound examinations. However, during the study, many participants expressed interest in undergoing additional ultrasound scans. In response, these scans were offered on a voluntary basis, with participants explicitly informed of their right to decline sharing scan data, even after undergoing the examination. The study strictly adhered to the ethical principles outlined in the Declaration of Helsinki (2013 revision; for further details, please refer to the official website of the World Medical Association at www.wma.net.). Adherence to these ethical principles was crucial throughout the research process, ensuring the protection of participants' rights, privacy, and safety. Additionally, a CONSORT flow diagram (S1 Fig) illustrates the participant recruitment, screening, eligibility assessment, and final allocation process.

### Data collection

Medical histories of the participants were collected through in-person interviews, supplemented by additional data gathered during subsequent telephone follow-ups. Trained nurses at the physical examination center of Taizhou People's Hospital performed anthropometric assessments, including measurements of weight (W), height (H), waist circumference (WC), and hip circumference (HC). Body mass index (BMI) was calculated by dividing weight by the square of height (W/H$^2$), and

waist-to-hip ratio (WHR) was calculated by dividing waist circumference by hip circumference (WC/HC). After a 15-minute rest period, blood pressure readings were taken twice from the right upper arm, and the average values for systolic blood pressure (SBP) and diastolic blood pressure (DBP) were recorded.

Venous blood samples were obtained following a fasting period exceeding 10 hours. At the laboratory of Taizhou People's Hospital, serum levels of various biomarkers were analyzed using a fully automated analyzer (Beckman Coulter AU5800, Brea, USA). These biomarkers encompassed glutamic-pyruvic transaminase (ALT), aspartate aminotransferase (AST), creatinine (Cr), uric acid (UA), FBG, total cholesterol (TC), triglycerides (TG), HDL, low-density lipoprotein choles-terol (LDL), and high-sensitive C-reactive protein (hs-CRP). HbA1c levels were measured using high-performance liquid chromatography with a TOSOH HLC-723G8 analyzer (Tokyo, Japan). Serum insulin (INS) and CP concentrations were quantified employing chemiluminescence particle immunoassay kits supplied by Abbott GmbH (Germany), with catalog numbers 3L53/09p36 and 8K41/04T75, respectively. LPCAT3 concentrations in serum were quantified using a commercial sandwich ELISA kit (JL21306, *JiangLai* Biotechnology, Shanghai, China) in accordance with the manufacturer's protocol. Serum samples were diluted 1:4 with using the buffer provided with the kit to ensure that the concentrations fell within the assay's linear range. The ELISA was conducted following standardized procedures, with absorbance measured at 450 nm using a microplate reader. Quantification relied on a standard curve spanning 0–40 ng/ml. The assay's performance was validated by confirming the kit's detection range (0.625–40 ng/mL) and sensitivity (0.29 ng/mL), ensuring accurate and reliable measurement of serum LPCAT3 levels. Insulin resistance (IR) was evaluated using the HOMA-IR formula: HOMA-IR = (INS * FBG)/ 22.5 [14]. Participants were instructed to follow a mixed diet containing a minimum of 150 grams of carbohydrates for three days preceding the OGTT. On the day of the OGTT, each participant consumed 75 grams of glucose dissolved in water within five minutes. Subsequently, a venous blood sample was collected two hours after the initiation of glucose ingestion to assess the blood glucose level.

Three ultrasound specialists conducted liver and carotid artery ultrasound scans and independently assessed the findings. The key diagnostic features of fatty liver encompass an enhanced echo of the liver parenchyma and decreased clarity of the intrahepatic vascular and ductal structures [15]. A carotid artery intima-media thickness (IMT) of ≥1.0 mm indicated intimal thickening, while localized thickening exceeding 1.5 mm was classified as plaque formation [16]. Any discrepancies identified during these assessments were resolved through group discussions among the three specialists.

## Data analysis

The sample size was estimated using PASS 15, with the significance level (α) set at 0.05, a target power (1 − β) of 0.8, and accounting for an expected dropout rate of 20%, resulting in a minimum required sample size of 318. To determine the normality of continuous variables, the Kolmogorov-Smirnov test was employed. These variables were subsequently presented as either mean ± standard deviation (SD) or median with the 25th percentile (P25) and 75th percentile (P75), depending on their distribution characteristics. For two-group comparisons, the T-test was used for normally distributed data, and the Mann-Whitney U test for non-normally distributed data. For three or more groups, One-Way ANOVA was applied to normally distributed data with homogeneous variances, and the Kruskal-Wallis H test otherwise. Categorical variables were compared using the chi-square test. Spearman correlation analysis was applied to explore the relation-ship between serum LPCAT3 levels and other indicators. Stepwise linear regression was employed to identify the inde-pendent factors influencing serum LPCAT3 levels. Variance inflation factors (VIFs) were calculated to assess collinearity among the independent variables. A VIF greater than 10 is deemed to signify severe collinearity. When high VIFs revealed significant collinearity, we explored data transformation methods like centering or standardizing the data. If there was a theoretical basis for an interaction between two variables, we considered adding interaction terms. Moreover, in complex cases where other methods were ineffective in addressing multicollinearity, we used principal component analysis (PCA) and partial least squares (PLS) Regression as dimensionality-reduction techniques. To develop the optimal regression models, we employed a stepwise approach to include or exclude variables, ensuring only statistically significant predictors

were retained. This process inherently considered the potential influence of multiple confounding factors. To refine model selection and balance model fit with complexity, we used statistical criteria like $R^2$, AIC, and BIC, which helped evaluate models while accounting for the impact of these confounders. Logistic regression analysis was conducted to investigate the association between LPCAT3 and the incidence of T2DM. To control for potential confounding factors, we considered several important variables that might influence both serum LPCAT3 levels and the incidence of T2DM, such as age, gender, BMI, blood pressure, and blood lipid levels. In the logistic regression analysis, these potential confounding factors were included as covariates in the models to adjust for their effects. Before performing the regression analyses, skewed data were log-transformed using the natural logarithm (base e) to achieve normality. To eliminate scale differences among variables and enhance model stability, all variables were standardized (Z-score normalization with mean = 0 and standard deviation = 1) prior to PLS analysis. To evaluate the predictive ability of serum LPCAT3 levels for T2DM onset, we generated a receiver-operating characteristic (ROC) curve. The curve plots sensitivity (true-positive rate) against 1-specificity (false-positive rate) across different LPCAT3 thresholds. We used the Youden index to find the optimal cut-off point, which balances sensitivity and specificity. The area under the curve (AUC) was calculated via the trapezoidal rule, and its 95% confidence interval was estimated using the DeLong method. ROC curves were generated using MedCalc version 20.0.14. PLS analysis was conducted with SIMCA 14.1, while other statistical analyses were performed using SPSS 22.0. A two-sided p-value < 0.05 was considered statistically significant.

## Results

### Comparison of clinical parameters and serum LPCAT3 levels between patients with T2DM and individuals with NGT

As presented in Table 1, individuals with T2DM exhibited higher levels of WC, WHR, BMI, SBP, ALT, eGFR, TG, FBG, 2hPG, HbA1c, HOMA-IR, and hs-CRP compared to those with NGT. In contrast, the T2DM group had lower levels of serum Cr and HDL. No significant differences were observed in gender distribution, age, DBP, AST, UA, TC, or LDL between the two groups. The median serum LPCAT3 level in the NGT group was 24.43 ng/ml, with an interquartile range of 14.41 to 49.37 ng/ml, which was higher than the median level of 21.51 ng/ml (interquartile range: 8.47 to 35.63 ng/ml) observed in the T2DM group.

### Correlation analysis between liver and carotid artery ultrasound assessments and serum LPCAT3 levels in NGT and T2DM groups

As illustrated in S1 Table, among all participants, 91.1% underwent liver ultrasound evaluations, and 69.3% completed carotid ultrasound assessments. These percentages were calculated based on the subset of participants who agreed to share their results, after excluding two individuals from each group (NGT and T2DM) who had declined consent. In the NGT group, 225 participants completed liver ultrasound scans. Among them, 71 were diagnosed with fatty liver while 154 were not. Concurrently, 135 participants in the NGT group completed carotid ultrasound assessments, with 26 individuals identified with carotid atherosclerosis and 109 without. In the T2DM group, 238 participants underwent liver scans. Of these, 160 had fatty liver and 78 did not. Additionally, 217 participants in the T2DM group completed carotid ultrasound assessments, revealing that 68 had carotid atherosclerosis and 149 did not. For both the NGT and T2DM groups, no statistically significant difference in serum LPCAT3 levels was observed between the subgroups with and without fatty liver, as well as between those with and without carotid atherosclerosis.

### Assessment of the associations between LPCAT3 levels and clinical parameters in the NGT and T2DM groups using Spearman correlation analysis

Table 2 reveals that in individuals with NGT, negative correlations were observed between LPCAT3 levels and HDL, FBG, and HbA1c. Additionally, among patients with T2DM, LPCAT3 levels showed negative associations with BMI and WC.

**Table 1. Basic characteristics of participants.**

| Variables | NGT | T2DM | $X^2$ or $z$ or $t$ value | $p$ value | Reference range* |
|---|---|---|---|---|---|
| N (male/female) | 256 (127/129) | 252 (132/120) | 0.39 | 0.53 | – |
| Age (year) | 51.07 ± 11.96 | 51.94 ± 11.47 | −0.84 | 0.40 | – |
| BMI (kg/m2) | 24.04 ± 3.21 | 25.48 ± 3.34 | −4.96 | <0.01 | 18.5-23.9 [T1] |
| WC (cm) | 87.86 ± 7.23 | 94.27 ± 9.11 | −8.79 | <0.01 | male: <85; female: <80 [T1] |
| WHR | 0.89 ± 0.05 | 0.94 ± 0.05 | −9.92 | <0.01 | male: <0.9; female: <0.85 [T1] |
| SBP (mmHg) | 128.23 ± 14.48 | 133.40 ± 16.44 | −3.76 | <0.01 | <140 [T2] |
| DBP (mmHg) | 81.41 ± 10.61 | 82.34 ± 9.99 | −1.01 | 0.31 | <90 [T2] |
| ALT (U/L) | 18.50 (13.25, 28.00) | 22.00 (15.25, 34.75) | −3.26 | <0.01 | 7-50 [T3] |
| AST (U/L) | 20.00 (17.00, 25.00) | 20.00 (17.00, 26.00) | −0.40 | 0.69 | 13-40 [T3] |
| Cr (umol/L) | 63.97 ± 14.97 | 58.32 ± 14.40 | 4.34 | <0.01 | male:57–111; female: 41–81 [T3] |
| UA (umol/L) | 332.03 ± 86.78 | 330.46 ± 93.67 | 0.20 | 0.84 | male:208–416; female:149–358 [T3] |
| eGFR (ml/min) | 102.47 ± 13.98 | 107.22 ± 14.94 | −3.70 | <0.01 | 90-120 [T3] |
| TC (mmol/L) | 4.78 ± 1.01 | 4.70 ± 1.14 | 0.78 | 0.44 | <5.2 [T3] |
| TG (mmol/L) | 1.28 (0.91, 1.97) | 1.55 (1.07, 2.57) | −3.55 | <0.01 | <1.7 [T3] |
| HDL (mmol/L) | 1.26 ± 0.29 | 1.17 ± 0.26 | 3.61 | <0.01 | >1.0 [T3] |
| LDL (mmol/L) | 3.00 ± 0.73 | 3.05 ± 0.82 | −0.70 | 0.48 | <3.4 [T3] |
| FBG (mmol/L) | 4.90 (4.47, 5.16) | 7.58 (6.18, 9.88) | −16.57 | <0.01 | 3.9-6.1 [T3] |
| 2hPG (mmol/L) | 6.36 (5.80, 6.91) | 12.17 (10.40, 14.96) | −19.45 | <0.01 | <7.8 [T3] |
| HbA1c (%) | 5.47 (5.32, 5.59) | 7.84 (7.04, 9.33) | −19.50 | <0.01 | 4.0-6.0 [T3] |
| HOMA-IR | 1.77 (1.22, 2.71) | 3.28 (1.91, 5.80) | −9.76 | <0.01 | – |
| hs-CRP (mg/L) | 0.56 (0.32, 1.00) | 1.20 (0.51, 2.06) | −6.73 | <0.01 | 0-3 [T3] |
| LPCAT3 (ng/ml) | 24.43 (14.41, 49.37) | 21.51 (8.47, 35.63) | −3.13 | <0.01 | – |

Continuous variables following a normal distribution are expressed as the mean ± standard deviation, with inter-group comparisons conducted using the t-test. For continuous variables that do not following a normal distribution, they are presented as the median (25th-75th percentiles), and inter-group comparisons are performed using the Mann-Whitney U test. Categorical variables are compared using the Chi-square test. A p-value of less than 0.05 is considered statistically significant. * The reference ranges were established in compliance with Chinese clinical standards. Specifically, Reference T1 followed the 'Guidelines for Medical Nutrition Treatment of Overweight/Obesity in China' (https://doi.org/10.6133/apjcn.202209_31(3).0013), Reference T2 was based on the 'Clinical Practice Guideline for the Management of Hypertension in China' (https://doi.org/10.1097/cm9.0000000000003431), and Reference T3 adhered to the 'Standard for Reference Intervals of Common Clinical Biochemistry Tests' (Standard No.: WS/T 404), issued by the National Health Commission of China. Additional details are available on the official website at http://www.nhc.gov.cn/. Abbreviations: NGT: normal glucose tolerance; T2DM: type 2 diabetes mellitus; BMI: body mass index; WC: waist circumference; WHR: waist-to-hip ratio; SBP: systolic blood pressure; DBP: diastolic blood pressure; ALT: Alanine aminotransferase; AST: Aspartate aminotransferase; Cr: creatinine; UA: uric acid; eGFR: estimated glomerular filtration rate; TC: total cholesterol; TG: triglyceride; HDL: high-density lipoprotein cholesterol; LDL: low density lipoprotein cholesterol; FBG: fasting blood glucose; 2hPG: 2-hour post-oral glucose tolerance test blood glucose level; HbA1c: glycated hemoglobin A1c; HOMA-IR: homeostasis model assessment of insulin resistance; hs-CRP: high sensitive C-reactive protein; LPCAT3: lysophosphatidylcholine acyltransferase 3.

## Stepwise linear regression analysis identifies independent factors influencing LPCAT3 levels

A stepwise linear regression analysis was performed, utilizing LPCAT3 as the dependent variable and the remaining variables as independent variables. As detailed in Table 3, the findings reveal that BMI, HDL, and FBG were identified as independent negative predictors of serum LPCAT3 levels. The model yielded an R-squared value of 0.049, indicating that it could only explain 4.9% of the variation in serum LPCAT3 levels. This suggests the existence of numerous unidentified factors that may contribute to the regulation of serum LPCAT3.

When assessing the impact of confounding factors on our regression models, we found that variables like age, gender, blood pressure, and liver/kidney function indicators generally had minimal effects on the outcomes (S2 Table).

**Table 2. Exploring the Spearman correlation between LPCAT3 and clinical parameters.**

| LPCAT3 | NGT | | T2DM | |
|---|---|---|---|---|
| | *r* | *p* | *r* | *p* |
| sex | −0.065 | 0.298 | 0.093 | 0.143 |
| age | −0.016 | 0.794 | 0.029 | 0.645 |
| BMI | −0.070 | 0.267 | −0.162 | <0.05 |
| WC | −0.037 | 0.551 | −0.153 | <0.05 |
| WHR | −0.029 | 0.643 | −0.058 | 0.361 |
| SBP | 0.001 | 0.992 | −0.043 | 0.501 |
| DBP | 0.033 | 0.599 | −0.105 | 0.097 |
| ALT | 0.019 | 0.762 | −0.109 | 0.085 |
| AST | 0.022 | 0.723 | −0.089 | 0.158 |
| Cr | 0.042 | 0.508 | −0.028 | 0.662 |
| UA | −0.028 | 0.656 | 0.016 | 0.800 |
| eGFR | 0.000 | 0.995 | −0.032 | 0.613 |
| TC | −0.093 | 0.136 | −0.019 | 0.764 |
| TG | 0.060 | 0.339 | −0.026 | 0.685 |
| HDL | −0.145 | <0.05 | −0.051 | 0.419 |
| LDL | −0.077 | 0.217 | −0.002 | 0.969 |
| FBG | −0.133 | <0.05 | −0.093 | 0.143 |
| 2hPG | −0.027 | 0.673 | −0.011 | 0.856 |
| HbA1c | −0.157 | <0.05 | 0.019 | 0.766 |
| HOMA-IR | −0.093 | 0.136 | −0.058 | 0.357 |
| hs-CRP | 0.027 | 0.672 | −0.042 | 0.507 |

Spearman correlation analysis was performed to evaluate the associations between LPCAT3 and clinical parameters. The results are presented as correlation coefficients (r) and their corresponding p-values. A p-value of less than 0.05 was considered statistically significant. Abbreviations: LPCAT3: lysophosphatidylcholine acyltransferase 3; NGT: normal glucose tolerance; T2DM: type 2 diabetes mellitus; BMI: body mass index; WC: waist circumference; WHR: waist-to-hip ratio; SBP: systolic blood pressure; DBP: diastolic blood pressure; ALT: Alanine aminotransferase; AST: Aspartate aminotransferase; Cr: creatinine; UA: uric acid; eGFR: estimated glomerular filtration rate; TC: total cholesterol; TG: triglyceride; HDL: high-density lipoprotein cholesterol; LDL: low density lipoprotein cholesterol; FBG: fasting blood glucose; 2hPG: 2-hour post-oral glucose tolerance test blood glucose level; HbA1c: glycated hemoglobin A1c; HOMA-IR: homeostasis model assessment of insulin resistance; hs-CRP: high sensitive C-reactive protein.

**Table 3. Independent predictors of serum LPCAT3 identified by stepwise linear regression.**

| Variables | unstandardised coefficients | | *t* | *p* | 95% CI for *β* | |
|---|---|---|---|---|---|---|
| | *β* | Std. Error | | | lower | upper |
| Constant | 5.106 | 0.440 | 11.613 | <0.01 | 4.242 | 5.969 |
| BMI | −0.037 | 0.013 | −2.859 | <0.01 | −0.063 | −0.012 |
| HDL | −0.390 | 0.154 | −2.531 | <0.05 | −0.693 | −0.087 |
| FBG | −0.393 | 0.121 | −3.244 | <0.01 | −0.630 | −0.155 |

Stepwise linear regression analysis was employed to explore the independent correlates of LPCAT3. The analysis results are presented as unstandardized coefficients, t-values, p-values, and 95% confidence intervals. A p-value less than 0.05 was considered statistically significant, indicating a significant association between the corresponding variable and LPCAT3. The R-squared value of this model is 0.049. Prior to correlation analysis, LPCAT3 and FBG were logarithmically transformed. Abbreviations: LPCAT3: lysophosphatidylcholine acyltransferase 3; CI: confidence interval; BMI: body mass index; HDL: high-density lipoprotein cholesterol; FBG: fasting blood glucose.

Notably, significant collinearity was observed among eGFR, Cr, and age. After evaluating each variable's contribution, we excluded eGFR from the model, as it provided no additional information beyond what was already captured by the other correlated variables. In contrast, obesity-related anthropometric measures, along with lipid and glucose metabolism indicators, substantially impacted the model's stability. This highlights the importance of these factors in our analysis.

Our analysis of the relationships between LPCAT3 and obesity-related anthropometric measures revealed a complex interplay. Specifically, although WHR alone showed statistical significance as an independent variable, its association was attenuated when considered alongside BMI or WC. When BMI and WC were separately included in the model, appropriate models were successfully fitted. Optimal model-fitting analysis revealed that the model incorporating BMI outperformed the one with WC. However, when both variables (BMI and WC) were simultaneously included in the model, despite an acceptable level of collinearity, neither achieved statistical significance (S3–S7 Tables). This underscores the delicate nature and context-dependency of these relationships, implying that the choice of anthropometric measures could potentially have a substantial impact on the outcomes.

In the context of lipid-related factors, TC and LDL exhibited significant collinearity. As conventional methods such as centering and standardization failed to produce satisfactory results, we applied PCA to TC and LDL. This strategy led to the development of a model with a relatively good fit. Nevertheless, our findings revealed that HDL lost its statistical significance when all lipid-related factors—including TC, LDL, and TG—were taken into account in the model (S8 Table). This suggests that the relationship between LPCAT3 and lipid-related factors is significantly more complex than initially hypothesized, transcending a mere negative correlation with HDL. Instead, it involves an intricate web of interactions among various lipid components, which collectively modulate LPCAT3 levels.

The relationship between glycometabolic indicators and LPCAT3 levels is also intricate. The linear regression model incorporating FBG outperforms others, as evidenced by a higher $R^2$ value, indicating a superior fit. However, when 2hPG, HbA1c, or HOMA-IR is substituted for FBG, each variable still leads to a statistically valid model, albeit with a reduced $R^2$ value. Notably, once FBG is included in the model, adding either 2hPG or HbA1c causes FBG to lose its statistical significance (S9–S14 Tables). This implies that while LPCAT3 may exhibit an association with various glycometabolic indicators, the relationships among them are intricate and interdependent. It is likely that the relatively stronger correlation observed between FBG and LPCAT3 may obscure the more subtle associations of the other indicators.

Overall, our analysis of collinearity among the independent variables has revealed a complex and unstable network of relationships which are highly dependent on the specific correlated variables incorporated into the model.

### PLS analysis of correlations between LPCAT3 and obesity-related anthropometric, lipid, and glucose indicators

By conducting collinearity and confounding factor analyses, we identified complex associations between LPCAT3 and various indicators. The primary focus was on obesity-related anthropometric indicators (e.g., BMI, WC, WHR), blood lipid indicators (e.g., TC, TG, HDL, LDL), and blood glucose indicators (e.g., FBG, 2hPG, HbA1c, HOMA-IR). To gain a deeper insight into these relationships, we performed partial least squares (PLS) analysis. In this analysis, LPCAT3 was designated as the dependent variable (Y), while the obesity-related anthropometric indicators, blood lipid indicators, and blood glucose indicators were treated as distinct sets of independent variables (X).

Fig 1A and 1B illustrate the PLS analysis results concerning the correlations between LPCAT3 and obesity-related anthropometric indicators. Fig 1A, a PLS bi-plot, shows that principal component 1 (PC1) explains approximately 78.6% of the total variance, while principal component 2 (PC2) accounts for about 18.7%. BMI, WC, and WHR are projected onto this bi-plot, contributing to the variation in the principal components. Notably, LPCAT3 exhibits a relatively strong negative correlation with WC and BMI. Fig 1B, a Coefficient Plot, displays coefficients of approximately −0.064 (SE = 0.186) for BMI, −0.067 (SE = 0.346) for WC, and −0.040 (SE = 0.207) for WHR. These coefficients suggest negative correlations between

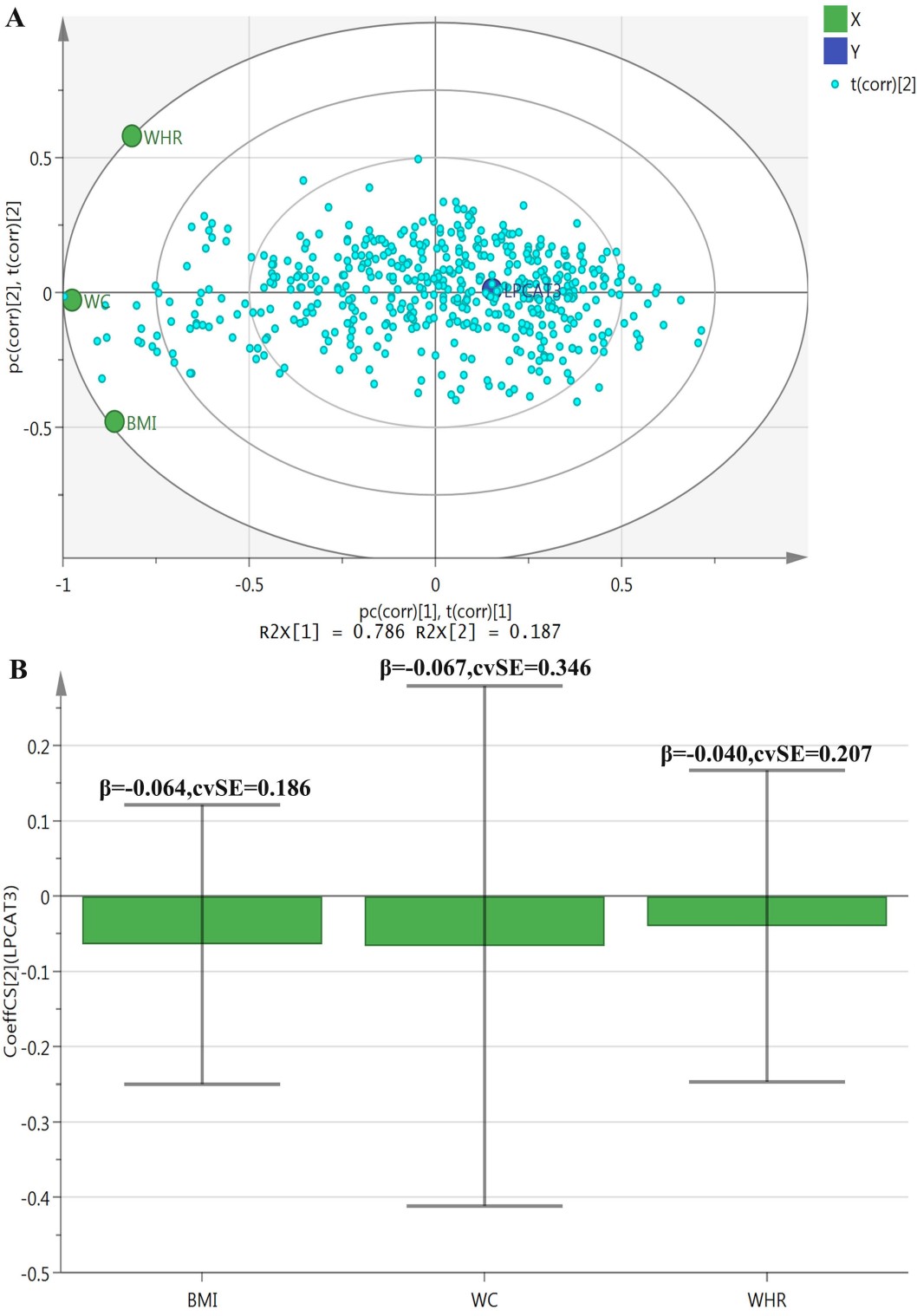

**Fig 1. PLS analysis of correlations between LPCAT3 and obesity-related anthropometric indicators.** In Panel A (PLS bi-plot), principal component 1 accounts for 78.6% and principal component 2 for 18.7% of total variance. BMI, WC, and WHR are projected in the plot, with LPCAT3 showing a relatively strong negative correlation with WC and BMI. Panel B (Coefficient Plot) presents correlation coefficients between LPCAT3 and obesity-related indicators: approximately −0.064 (SE = 0.186) for BMI, −0.067 (SE = 0.346) for WC, and −0.040 (SE = 0.207) for WHR. All variables were standardized

prior to PLS analysis. Abbreviations: PLS: partial least squares; LPCAT3: lysophosphatidylcholine acyltransferase 3; BMI: body mass index; WC: waist circumference; WHR: waist-to-hip ratio.

all three indicators and LPCAT3, with BMI and WC potentially having a stronger impact. Nevertheless, the relatively large standard errors indicate low stability and reliability of these estimates, reflecting considerable uncertainty in the observed correlations.

Fig 2A and 2B present the PLS analysis results concerning the correlations between LPCAT3 and blood lipid indicators. Fig 2A, a PLS bi-plot, shows $R^2$ values of $R^2X[1] = 0.558$ and $R^2X[2] = 0.203$ for the first two components, indicating the proportion of variance in the X-variables explained by each component. TC, HDL, and LDL demonstrate certain correlations with the principal components, with HDL exhibiting a relatively strong negative correlation with LPCAT3. In contrast, TG contributes less to the variation in the principal components and shows a weaker association with LPCAT3. Fig 2B, a Coefficient Plot, displays coefficients of approximately −0.001 (SE = 0.087) for TC, −0.017 (SE = 0.185) for TG, −0.071 (SE = 0.219) for HDL, and −0.009 (SE = 0.106) for LDL. HDL appears to have a slightly stronger negative correlation with LPCAT3 compared to the other lipid indicators. However, the relatively large standard errors suggest imprecise model estimates, thereby limiting the reliability of the findings.

Fig 3A and 3B depict the PLS analysis results regarding the correlations between LPCAT3 and blood glucose indicators. Fig 3A, a PLS bi-plot, shows $R^2$ values of $R^2X[1] = 0.783$ and $R^2X[2] = 0.098$ for the first two components. FBG, 2hPG, HbA1c, and HOMA-IR are projected onto this bi-plot, contributing to the variation in the principal components. FBG shows the strongest negative correlation with LPCAT3, followed by HOMA-IR. Fig 3B, a Coefficient Plot, presents coefficients of approximately −0.115 (SE = 0.213) for FBG, −0.023 (SE = 0.148) for 2hPG, 0.020 (SE = 0.225) for HbA1c, and −0.069 (SE = 0.365) for HOMA-IR. FBG and HOMA-IR appear to have relatively stronger negative correlations with LPCAT3. Still, the relatively large standard errors indicate low stability and reliability of these coefficient estimates, introducing a degree of uncertainty into the observed correlations.

In summary, the PLS analysis highlights the intricate and multifaceted associations among these variables. These associations collectively influence LPCAT3 levels. Although correlations between LPCAT3 and the examined indicators (obesity-related anthropometric, blood lipid, and blood glucose) are observed, these correlations are notably weak and exhibit considerable uncertainty, as indicated by the relatively large standard errors. This underscores the complexity of these relationships and the need for cautious interpretation of the results.

S2 Fig displays the prediction versus observation plots for the three models. Generally, the Root Mean Squared Error of Estimation (RMSEE) and Root Mean Squared Error of Cross-Validation (RMSECV) values of these three models are quite similar, suggesting that all three models exhibit stability and good generalizability. Nevertheless, the comparatively high values of RMSEE and RMSECV indicate that the predictive performance of these models is unsatisfactory.

**Logistic regression analysis of T2DM incidence across serum LPCAT3 tertiles, using the lowest tertile as reference**

The 506 participants were categorized into three groups based on their serum LPCAT3 concentrations: T1 (< 15.965 ng/mL), T2 (15.965–32.614 ng/mL), and T3 (> 32.614 ng/mL). The incidence of T2DM showed a progressive decline with increasing LPCAT3 levels, with rates of 55.88% in T1, 51.48% in T2, and 41.42% in T3. As detailed in Table 4, the initial analysis revealed a significantly lower incidence of T2DM in the T3 group compared to the T1 group. However, after sequentially adjusting for obesity-related anthropometric indicators including BMI, WC and WHR, the difference between these two groups became insignificant. No statistical difference in T2DM incidence was observed between the T1 and T2 groups.

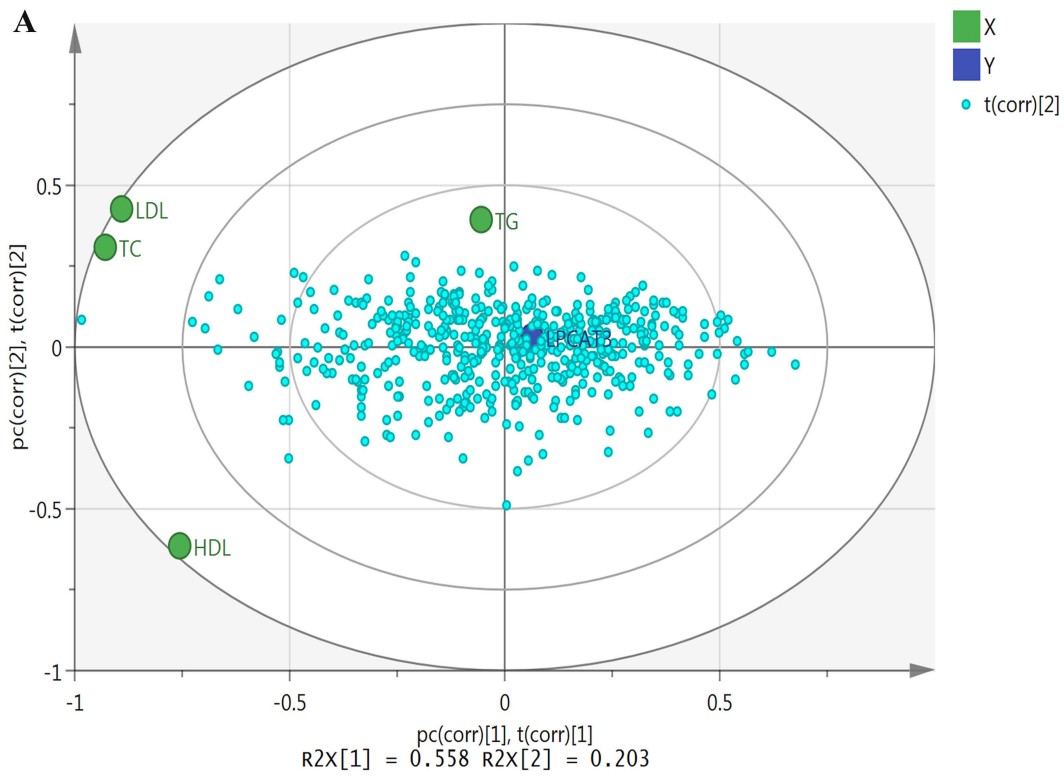

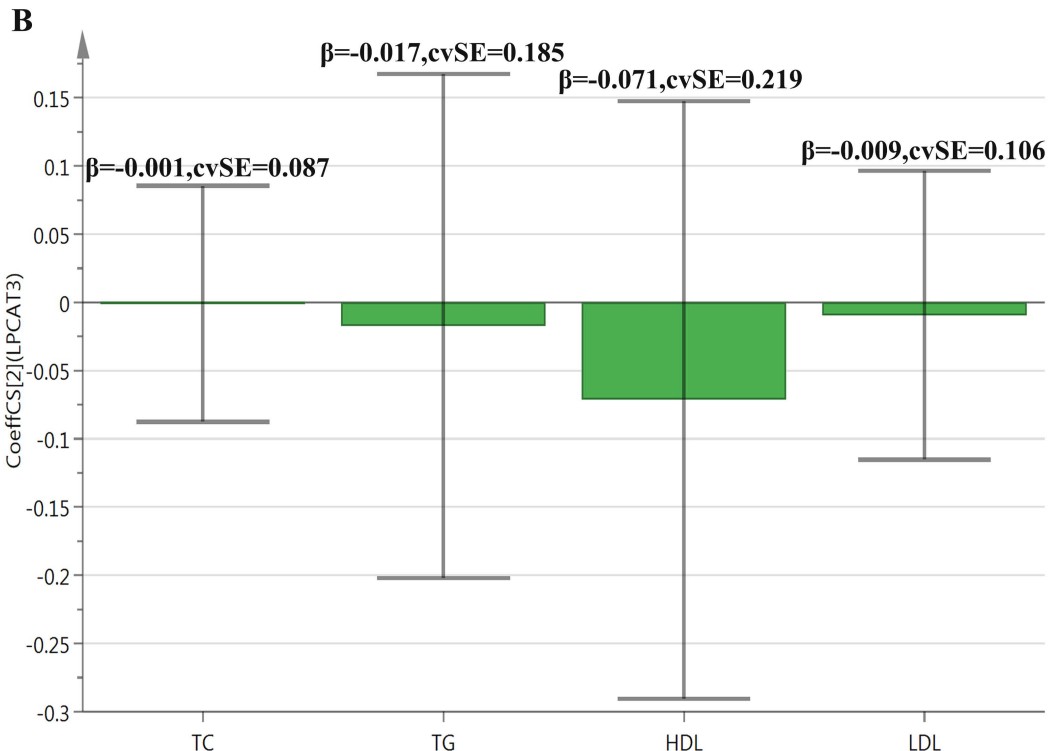

**Fig 2. PLS analysis of correlations between LPCAT3 and blood glucose indicators.** Panel A (PLS bi-plot) displays R² values of R²X[1] = 0.558 and R²X[2] = 0.203, indicating the proportion of variance in X-variables explained by each component. TC, TG, HDL, and LDL correlate with LPCAT3, with HDL showing a relatively strong negative correlation and TG a weaker one. Panel B (Coefficient Plot) shows correlation coefficients between LPCAT3

and blood lipid indicators: about −0.001 (SE = 0.087) for TC, −0.017 (SE = 0.185) for TG, −0.071 (SE = 0.219) for HDL, and −0.009 (SE = 0.106) for LDL. All variables were standardized prior to PLS analysis. Abbreviations: PLS: partial least squares; LPCAT3: lysophosphatidylcholine acyltransferase 3; TC: total cholesterol; TG: triglyceride; HDL: high-density lipoprotein cholesterol; LDL: low density lipoprotein cholesterol.

### ROC curve analysis: evaluating serum LPCAT3 as a biomarker for predicting T2DM incidence via AUC

Fig 4 demonstrates that serum LPCAT3 levels have a statistically significant yet relatively weak predictive capability for the incidence of T2DM. S15 Table presents the results of an ROC curve analysis for predicting T2DM incidence based on serum LPCAT3 levels. The optimal cut-off point for LPCAT3 was determined to be 30.133 ng/ml, with a sensitivity of 42.19% (95% CI: 36.1–48.5) and a specificity of 71.83% (95% CI: 65.8–77.3). The Youden index at this cut-off point, calculated as Sensitivity + Specificity − 1, would be approximately 0.140. The area under the curve (AUC) was 0.580 (95%CI: 0.531–0.630, $p < 0.01$), indicating the overall discriminative ability of serum LPCAT3 levels in predicting T2DM incidence.

### Gender- and age-stratified regression analysis of the associations between LPCAT3, metabolic parameters (BMI, HDL, FBG), and T2DM risk

In metabolic disease research, sex and age are commonly recognized as key confounding factors that may disrupt the complex interrelationships among metabolic parameters and disease risks. Given this context, we conducted gender- and age-stratified regression analyses to gain a deeper understanding of the associations between LPCAT3, BMI, HDL, FBG, and the risk of T2DM.

In the gender-stratified analysis of metabolic parameters related to serum LPCAT3, we found significant negative correlations between LPCAT3 and BMI, HDL, and FBG in males. In females, only the correlation between LPCAT3 and BMI was significant, while no significant correlations were observed with HDL and FBG. The $R^2$ values for the male and female models were 0.089 and 0.029, respectively (S16 Table). These findings indicate stronger correlations between LPCAT3 and the parameters in males, implying gender-specific regulation of LPCAT3.

Subsequently, we divided the sample into three age groups (<40 years, 40–59 years, and ≥60 years) to further explore the age-specific associations. In the group aged <40 years, LPCAT3 showed no significant correlations with BMI, HDL, or FBG. In the group aged 40–59 years, a significant negative correlation was found between LPCAT3 and FBG. For the group aged ≥60 years, only BMI showed a significant negative correlation with LPCAT3, while no significant correlations were found between LPCAT3 and HDL or FBG. The $R^2$ values for the regression models corresponding to these three age groups were 0.052, 0.038, and 0.097, respectively (S17 Table). Taken together, these results suggest that the associations between LPCAT3 and metabolic parameters may vary across different age groups.

To gain a deeper understanding of the association between serum LPCAT3 levels and the risk of T2DM, we incorporated age and gender as key covariates into the analysis, building upon the binary logistic regression framework. Given that obesity-related measures may confound or mediate this association, BMI was also considered a key factor.

The initial binary logistic regression analysis, which only considered serum LPCAT3 levels as a predictor, revealed a negative association between LPCAT3 levels and the incidence of T2DM (S18 Table). However, when sex, age, BMI, and their interaction terms with LPCAT3 were incorporated into the model, no significant interaction effects between LPCAT3 and these variables were detected. Moreover, the inclusion of these factors attenuated the initially observed association between LPCAT3 levels and T2DM risk (S19 Table).

To gain deeper insights, we conducted further stratified analyses. In males, LPCAT3 emerged as a significant negative predictor of T2DM risk. A one-unit increase in logarithmically transformed LPCAT3 levels was associated with a 37.3% decrease in the odds of developing T2DM. In females, although the odds ratio indicated a decrease in risk, it was not statistically significant (S20 Table). In the age-stratified analysis, only the 40–59 age group showed a significant negative association between LPCAT3 and T2DM risk. A one-unit increase in logarithmically transformed LPCAT3

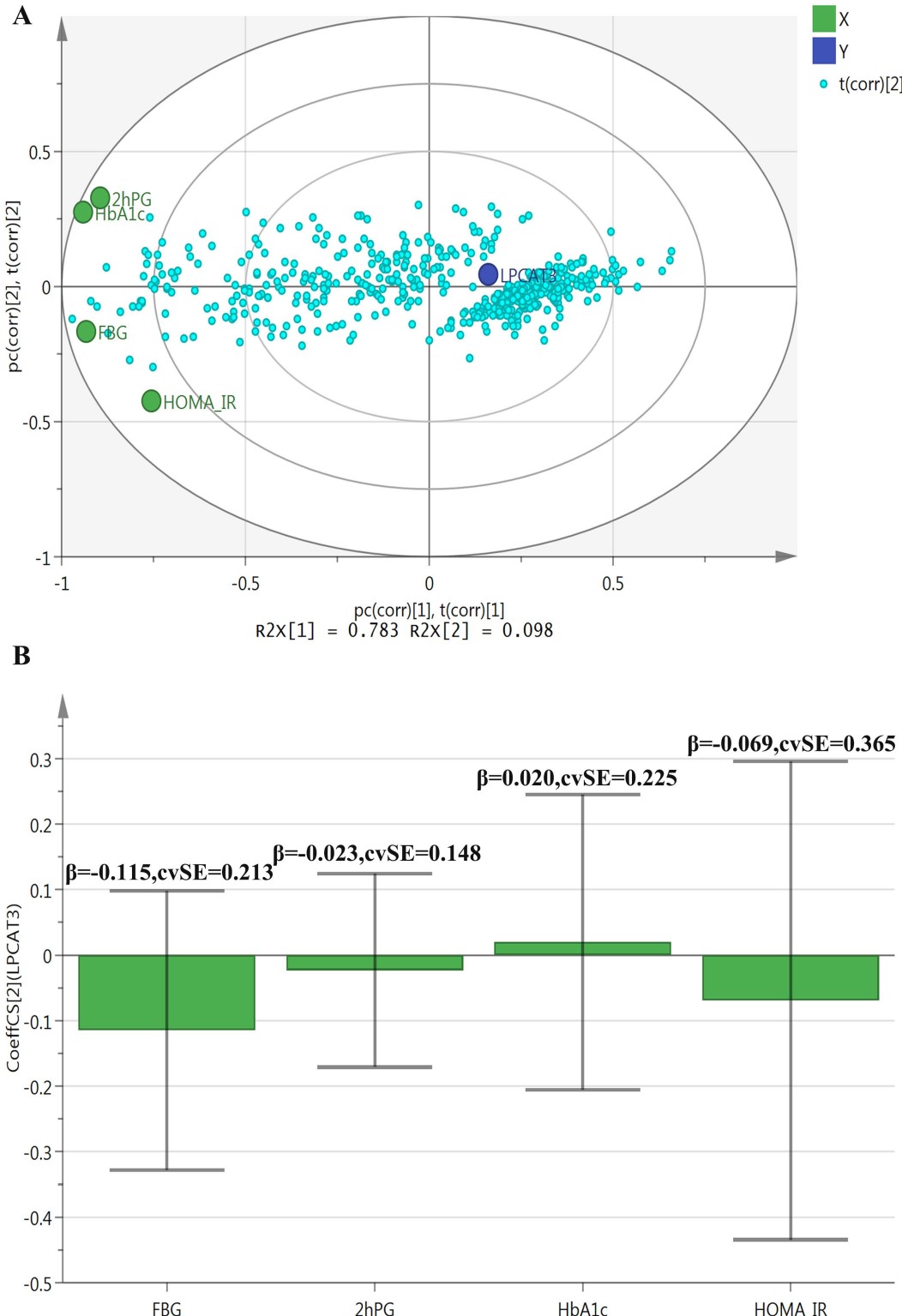

**Fig 3. PLS analysis of correlations between LPCAT3 and blood glucose indicators.** Panel A (PLS bi-plot) presents R² values of R²X[1] = 0.783 and R²X[2] = 0.098. FBG, 2hPG, HbA1c, and HOMA-IR are projected, with FBG showing the strongest negative correlation with LPCAT3, followed by HOMA-IR. Panel B (Coefficient Plot) presents correlation coefficients between LPCAT3 and blood glucose indicators: roughly −0.115 (SE = 0.213) for

FBG, −0.023 (SE = 0.148) for 2hPG, 0.020 (SE = 0.225) for HbA1c, and −0.069 (SE = 0.365) for HOMA-IR. All variables were standardized prior to PLS analysis. Abbreviations: PLS: partial least squares; LPCAT3: lysophosphatidylcholine acyltransferase 3; FBG: fasting blood glucose; 2hPG: 2-hour post-oral glucose tolerance test blood glucose level; HbA1c: glycated hemoglobin A1c; HOMA-IR: homeostasis model assessment of insulin resistance.

**Table 4. Logistic regression analysis of serum LPCAT3 level and T2DM prevalence.**

|  | T1 | T2 |  | T3 |  |
|---|---|---|---|---|---|
|  | reference | OR (95%CI) | p | OR (95%CI) | p |
| Model 1 | 1 | 0.838 (0.546, 1,284) | 0.416 | 0.558 (0.363, 0.859) | <0.01 |
| Model 2 | 1 | 0.830 (0.536, 1,284) | 0.402 | 0.564 (0.364, 0.873) | <0.05 |
| Model 3 | 1 | 0.760 (0.467, 1,237) | 0.270 | 0.613 (0.379, 0.991) | <0.05 |
| Model 4 | 1 | 0.799 (0.480, 1,330) | 0.389 | 0.579 (0.352, 0.953) | <0.05 |
| Model 5 | 1 | 0.885 (0.514, 1,524) | 0.658 | 0.651 (0.382, 1.111) | 0.115 |

Logistic regression analysis was conducted to evaluate the association between serum lysophosphatidylcholine acyltransferase 3 (LPCAT3) tertiles and the incidence of type 2 diabetes mellitus (T2DM). The results of the logistic regression analysis, including incidence rates, odds ratios (OR), 95% confidence intervals (CI), and p-values, are presented. A p-value of less than 0.05 was considered statistically significant, indicating a significant association between the variables. The analysis included the following models: Model 1: crude model (unadjusted); Model 2: adjusted for sex (with male as the reference category), age, systolic blood pressure (SBP) and diastolic blood pressure (DBP); Model 3: adjusted for the variables in model 2 plus alanine aminotransferase (ALT), aspartate aminotransferase (AST), creatinine (Cr), uric acid (UA), and high-sensitive C-reactive protein (hs-CRP); Model 4: adjusted for the variables in model 3 plus total cholesterol (TC), triglycerides (TG), high-density lipoprotein cholesterol (HDL), and low-density lipoprotein cholesterol (LDL); Model 5: adjusted for the variables in model 4 plus body mass index (BMI), waist circumference (WC), and waist-to-hip ratio (WHR). Prior to correlation analysis, ALT, AST, TG and hs-CRP were logarithmically transformed. Given the significant collinearity between TC and LDL, common factors extracted from these two variables were employed to replace them in the regression model.

levels was associated with a 32.9% decrease in the odds of developing T2DM (S21 Table). Furthermore, we conducted a BMI-stratified analysis. We used a cutoff of 24 kg/m$^2$, which is a commonly used threshold in China to distinguish normal weight from overweight/obesity. The analysis demonstrated that in the BMI ≥ 24 kg/m$^2$ group, serum LPCAT3 was a significant negative predictor of T2DM risk. Specifically, each one-unit increase in logarithmically transformed LPCAT3 was associated with a 33.3% reduction in the odds of developing T2DM. In contrast, in the BMI < 24 kg/m$^2$ group, the association between LPCAT3 and T2DM risk was not statistically significant (S22 Table). Overall, the negative association between LPCAT3 and T2DM risk varied across genders, age groups, and BMI levels, with significant findings mainly observed in males, the 40–59 age group, and the BMI ≥ 24 kg/m$^2$ group.

## Discussion

In the present study, we found that participants newly diagnosed with T2DM had lower serum levels of LPCAT3 compared to individuals with NGT. Within the NGT cohort, we observed a negative correlation between serum LPCAT3 concentrations and both FBG and HbA1c levels. Importantly, FBG was identified as a potential independent negative predictor of LPCAT3 levels. These findings collectively suggest a complex and multifaceted involvement of LPCAT3 in glucose metabolic pathways, warranting further investigation to elucidate the underlying mechanisms and potential therapeutic implications. In 2016, Cash JG et al. were the pioneers in reporting that the overexpression of LPCAT3 in the liver alleviated postprandial hyperglycemia and improved glucose tolerance. They elucidated that this beneficial effect stemmed from the reduction of intracellular lysophosphatidylcholine levels, consequently attenuating its inhibitory influence on mitochondrial fatty acid β-oxidation [17]. However, in 2023, Tian Y et al. discovered that liver-specific deletion of LPCAT3 reduced polyunsaturated phosphatidylcholine (PUFA-PC) content within liver cell membranes. This alteration in phospholipid composition may enhance insulin receptor (INSR) endocytosis and signaling by modifying the physical properties of the membrane. Notably, mice with suppressed LPCAT3 expression exhibited upregulated hepatic fibroblast growth factor

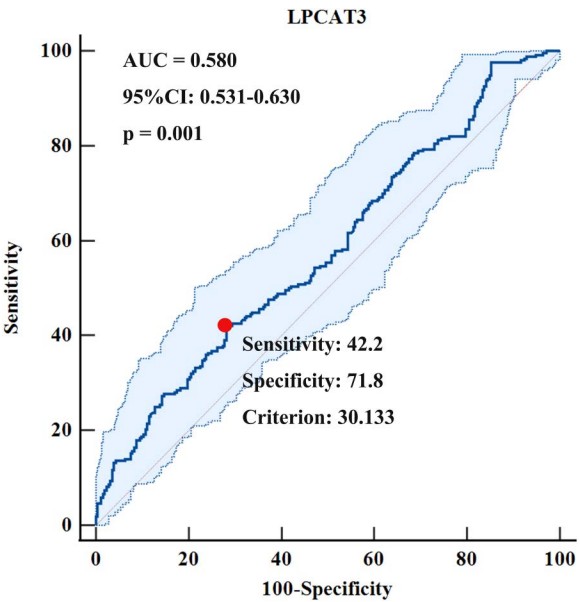

**Fig 4. ROC curve for predicting T2DM incidence using serum LPCAT3 levels.** The red dot in the figure represents the optimal cut-off point for LPCAT3, which is 30.133 ng/ml, with a sensitivity of 42.2% and a specificity of 71.8%. The area under the curve (AUC) was 0.580, and the 95% confidence interval for the AUC ranged from 0.531 to 0.630 (p < 0.01).

21 (FGF21) expression, increased glucose uptake in brown adipose tissue (BAT), and improved systemic insulin sensitivity and glucose tolerance [9]. Apart from the liver, targeted inhibition of LPCAT3 in skeletal muscle and adipose tissue has also been shown to enhance insulin signaling and improve glucose metabolism, consistent with the findings presented by Tian Y et al. In skeletal muscle, the suppression of LPCAT3 led to an enhancement in INSR phosphorylation, thereby augmenting insulin signaling and sensitivity. This effect was attributed to alterations in the lipid arrangement within the cell membrane, particularly through the promotion of Monosialotetrahexosylganglioside (GM-1) microdomain clustering. Mice with a skeletal muscle-specific deletion of LPCAT3 demonstrated increased insulin sensitivity and improved blood glucose regulation [10]. Conversely, the overexpression of LPCAT3 resulted in diminished insulin sensitivity [18]. Likewise, adipose-specific suppression of LPCAT3 decreased tissue PUFA-PC levels, concurrently increasing phosphorylation of INSR and AKT, thereby indicating the activation of the insulin signaling pathway. This activation subsequently enhanced insulin sensitivity and improved glucose tolerance [11]. In a murine model where systemic suppression of LPCAT3 expression was achieved through the administration of antisense oligonucleotides, significant improvements were observed in both glucose tolerance and insulin sensitivity [19]. In general, most studies support the idea that inhibiting LPCAT3 offers benefits in strengthening insulin signaling, alleviating insulin resistance, and improving glucose tolerance. Elevated activity and expression of LPCAT3 have also been noted in animal models exhibiting insulin resistance [9,11]. However, contrary to initial expectations, our research indicates that serum LPCAT3 levels are decreased in patients with T2DM and show a negative correlation with FBG. The underlying causes of this discrepancy remain elusive, prompting us to explore potential contributing factors. Notably, previous in vivo studies have primarily focused on elucidating how LPCAT3 expression in animal tissues affects glucose metabolism. In contrast, our study delves into the levels of LPCAT3 present in human serum. Therefore, it is of great significance to explore the source and function of serum LPCAT3. As we known, LPCAT3 is an endoplasmic reticulum (ER) membrane protein highly expressed in various metabolic tissues, including the liver, small intestine, and adipose tissue, as well as in other tissues such as the testis and kidney [20]. ER proteins are typically secreted via mechanisms such as co-translational translocation, post-translational translocation, and vesicular transport

[21]. Although no former studies have clarified the source of serum LPCAT3, it is inferred that it may be secreted through these ER pathways or released due to cell damage. Endoplasmic reticulum stress, inflammatory responses, lipid metabolism disorders, and cellular damage induced by T2DM [3] may contribute to these processes. It is important to note that serum LPCAT3 may not originate from a single tissue but rather from the cumulative contributions of multiple tissues expressing LPCAT3. The release of LPCAT3 from tissues into the serum is influenced not only by cellular secretory mechanisms but also by other factors, such as the efficiency of blood circulation and the metabolic clearance by the liver and kidneys. These factors may lead to serum LPCAT3 levels not accurately reflecting the actual tissue levels. Furthermore, systemic LPCAT3 is collectively modulated by multiple tissues and organs. This collective modulation potentially hinders a direct reflection of the biological functions of individual tissues through systemic LPCAT3. Additionally, the expression of LPCAT3 in specific tissues exhibits distinct regulatory patterns, and its biological functions vary across different tissues. For instance, in the liver, LPCAT3 is involved in lipogenesis and lipid transport and is regulated by the liver X receptor (LXR) [7]. In contrast, in macrophages, it is modulated by inflammatory signals and plays a role in affecting inflammation [22]. Given these tissue-specific regulatory patterns and functional differences, the functions of systemic LPCAT3 are unlikely to be a simple summation of those in individual tissues. In conclusion, LPCAT3 has a complex origin, modulation, and tissue-specific functions. Considering these characteristics, both the serum levels and functions of LPCAT3 may not fully represent its role in any single specific tissue. Therefore, further investigation is needed. Apart from considering the source of serum LPCAT3, it is imperative to take into account other possible factors that may affect serum LPCAT3. For instance, in patients with T2DM, persistent hyperglycemia and insulin resistance may trigger certain compensatory mechanisms, ultimately affecting the expression or activity of LPCAT3. Moreover, T2DM is a complex and multifaceted disorder, presenting with chronic inflammation and dysregulation of lipid metabolism, among other pathological features. The potential involvement of LPCAT3 in these processes will be discussed in subsequent sections. It is noteworthy that in our study, the association between LPCAT3 and T2DM, as well as glucose metabolism indicators, appears relatively weak. This is evidenced by the small area under the ROC curve, low R-squared values in the predictive models, and substantial standard errors from PLS analysis, which collectively highlight the tenuous and complex nature of this association. To gain a deeper understanding, future research should adopt a more comprehensive approach. Studies with larger sample sizes are essential to enhance statistical reliability. Additionally, examining LPCAT3 expression across a broader range of tissues may elucidate its overall role in glucose metabolism. Moreover, conducting long-term observations of the interaction between LPCAT3 and glucose metabolism over time will provide a clearer understanding of their relationship, rather than relying solely on one-time data.

The pivotal role of LPCAT3 in lipid metabolism has garnered substantial attention, emerging as a prominent topic of discussion. Its deficiency initiates a cascade of events that exert a profound influence on various lipid-related processes. Primarily, LPCAT3 deficiency activates β-catenin-dependent Wnt signaling pathway, which subsequently suppresses the expression of genes and transcription factors crucial for lipid synthesis, including fatty acid synthase (FAS), diacylglycerol O-acyltransferase homolog 1 (DGAT1), sterol regulatory element-binding protein (SREBP)-1c, SREBP-2, CCAAT/enhancer-binding protein alpha (C/EBPα), and peroxisome proliferator-activated receptor γ (PPARγ). Consequently, adipocyte differentiation and lipid accumulation are impaired [23]. Notably, among these factors, the downregulation of the SREBP-1c pathway due to LPCAT3 deficiency—a well-documented phenomenon in research—slows down fat production and accumulation [8,9,19,24–26]. Secondly, LPCAT3 is indispensable for lipid absorption, transport, secretion, and storage. As a key enzyme involved in phosphatidylcholine remodeling, LPCAT3 modulates the composition and fluidity of cell membrane phospholipids, thereby influencing lipid absorption and secretion. Studies on LPCAT3-knockout mice have shown a significant reduction in intestinal lipid absorption capacity, accompanied by decreased plasma levels of cholesterol, phospholipids, and triglycerides [27–29]. Furthermore, LPCAT3 promotes the formation of a phospholipid membrane microenvironment that is enriched in polyunsaturated fatty acids (PUFAs), particularly arachidonic acid. This microenvironment facilitates triglyceride accumulation within the membrane, enhancing triglyceride transport efficiency. In

the absence of LPCAT3, triglycerides cannot be effectively incorporated into lipoproteins, leading to their accumulation in liver and intestinal cells and a decrease in plasma triglyceride levels [30,31]. Moreover, the enzymatic activity of LPCAT3 on cellular membranes increases the production of ω-6 arachidonic acid-containing phosphatidylethanolamine, which promotes triglyceride storage within adipocytes [32]. Additionally, LPCAT3 affects the abundance of membrane proteins, including niemann-pick C1-like 1 (NPC1L1), ATP-binding cassette transporter A1 (ABCA1), and ATP-binding cassette transporter G8 (ABCG8), which are vital for lipid absorption and cholesterol secretion. LPCAT3 deficiency disrupts the intestinal absorption of cholesterol, triglycerides, and phospholipids and reduces the capacity of small intestinal epithelial cells to secrete cholesterol via chylomicron and HDL pathways [27]. Finally, LPCAT3 deficiency disrupts the assembly of very low-density lipoprotein (VLDL) particles, resulting in the formation of smaller VLDL particles with reduced lipid content. This underscores the critical role of LPCAT3 in VLDL biogenesis [31]. In conclusion, LPCAT3 is of paramount importance in lipid metabolism, and reduced LPCAT3 expression may be associated with lower levels of serum cholesterol and triglycerides. In this study, we investigated the associations between serum LPCAT3 levels and various lipid-related parameters in individuals with NGT and those with T2DM, uncovering several intriguing discoveries. Specifically, within the NGT group, we observed a negative correlation between serum LPCAT3 levels and HDL concentrations. Stepwise linear regression analysis subsequently identified HDL as an independent negative predictor of serum LPCAT3 levels. However, it is crucial to acknowledge that even within the NGT population, the correlation between LPCAT3 and HDL is relatively weak, as evidenced by the small correlation coefficient. In linear regression models, the attenuated association between HDL and LPCAT3 when adjusting for other lipid factors highlights the intricate regulatory network of lipid metabolism. PLS analysis further reveals the instability and complexity of the relationship between LPCAT3 and the overall lipid profile. Given that the current study does not fully elucidate these intricate connections, further research is warranted. Future research should aim to decipher the precise mechanisms underlying LPCAT3's interactions with lipid components within this complex network, potentially through large-scale longitudinal monitoring of LPCAT3 and lipid levels, as well as in vitro and in vivo manipulation of LPCAT3 activity.

Our study has uncovered a notable negative correlation between serum LPCAT3 levels and obesity-related body measurements, including BMI and WC. This finding appears to contradict previous researches that have focused on the tissue-specific expression or functional roles of LPCAT3. Previous studies have reported elevated LPCAT3 expression in the skeletal muscle of obese individuals [10,18,19,24]. Conversely, mice with liver-specific [9,19] or small intestine-specific [27] LPCAT3 deficiency exhibit reduced subcutaneous white adipose tissue, weight loss, and resistance to obesity. In addition to its roles in lipid biosynthesis, absorption, and transport (as previously discussed), LPCAT3 may also modulate energy metabolism, thereby influencing body weight regulation. In mice with reduced LPCAT3 activity, adipose tissue displays disrupted lipid metabolism, characterized by increased lipid hydrolysis and fatty acid cycling, which ultimately contributes to sustained energy expenditure and limited weight gain [32]. Moreover, a global knockout of LPCAT3 results in a decrease in the respiratory exchange ratio, indicating a shift towards using fat as the primary energy source. This shift may be associated with weight loss and reduced adipose tissue mass [19]. Additionally, inhibiting LPCAT3 may enhance the secretion of FGF21, which in turn increases energy expenditure [19]. While tissue-specific suppression of LPCAT3 expression has been shown to confer resistance to weight gain in preclinical models, our research reveals a potential negative trend between serum LPCAT3 levels and obesity-related anthropometric measures. This inconsistency implies a unique role of serum LPCAT3 distinct from its tissue-based function or suggests the presence of previously undiscovered mechanisms. In our study of individuals with T2DM, serum LPCAT3 levels exhibit a negative correlation with BMI and WC. Stepwise linear regression analysis indicates that BMI is an independent negative predictor of LPCAT3 levels. However, the strength of this association is relatively weak, as evidenced by the correlation coefficient. Subsequent multiple linear regression models and PLS analysis further reveal an unstable yet complex relationship between the three obesity-related anthropometric parameters included in this study (BMI, WC, and WHR) and serum LPCAT3 levels. To enhance our understanding of these relationships, future research should utilize advanced body composition assessment techniques, such

as dual-energy X-ray absorptiometry (DXA) scans, which provide detailed information on body fat distribution, to distinguish the contributions of visceral and subcutaneous adipose tissue depots. Furthermore, longitudinal studies tracking concurrent changes in anthropometric measures and serum LPCAT3 levels over time will advance our understanding of their interplay in metabolic health.

Notably, when serum LPCAT3 is used as a predictor for the onset of T2DM, its predictive power diminishes after adjusting for obesity-related anthropometric measures. A further stratified analysis based on body mass index (BMI) has shown that the negative correlation between serum LPCAT3 levels and T2DM is more pronounced in overweight/obese individuals (BMI ≥ 24 kg/m²). In contrast, this correlation did not reach statistical significance among those with a BMI < 24 kg/m². These observations highlight the intricate interplay among serum LPCAT3 levels, T2DM, and obesity. It is plausible that the association between LPCAT3 and T2DM is indirect, potentially mediated by obesity and adipocyte-related biological mechanisms. This hypothesis holds some merit. Firstly, obesity, a well-established risk factor for T2DM, manifests as insulin resistance, chronic low-grade inflammation, and dyslipidemia in affected individuals [33]. These metabolic disturbances may affect the physiological interaction between LPCAT3 and T2DM. Moreover, previous studies have shown that the regulatory effects of LPCAT3 on insulin signaling in various tissues, including the liver [9], skeletal muscle [10,18], and adipose tissue [11,17], are mediated by changes in membrane lipid composition and fluidity. Notably, obesity not only directly affects membrane lipid composition, leading to decreased fluidity [34], but also modulates LPCAT3 expression in the liver [24] and skeletal muscle [8,18], thereby influencing glycerophospholipid remodeling in cell membranes. Recognizing the importance of understanding the intricate associations among LPCAT3, obesity, and T2DM is paramount. Nevertheless, the present study falls short of further elucidating the intricate underlying mechanisms. Large-scale population-based cohort studies are needed to validate the LPCAT3-T2DM correlation and explore its variation across different obesity levels. Incorporating more confounding factors and using more specific animal or cellular models can also enhance our understanding of their relationship. These efforts may lead to novel therapeutic approaches for T2DM, especially in obese individuals.

Given that gender and age are crucial factors in the pathogenesis of metabolic diseases, we conducted stratified analyses to identify potential effect modifiers. Our findings indicate that serum LPCAT3 levels exhibit a more robust negative correlation with BMI, HDL, and FBG in males. To the best of our knowledge, no prior studies have reported gender-based disparities in LPCAT3 expression or function concerning metabolic disorders. However, it is widely acknowledged that males and females demonstrate significant differences in metabolic parameters and hormonal profiles [35]. We hypothesize that these gender-related differences may elucidate the observed variations in the associations between LPCAT3 levels and metabolic indicators. Interestingly, our study observed substantial disparities in multiple baseline parameters between males and females. For instance, males had higher BMI and lower HDL levels compared to females (S23 Table). Nevertheless, these baseline differences alone cannot fully elucidate the gender-specific variations in the associations between LPCAT3 levels and metabolic parameters. For example, in a further BMI-stratified analysis, we found no evidence to suggest that these associations could be solely attributed to BMI differences (S24 Table). This implies that the observed differences may arise from multiple confounding factors or other unidentified variables. Gender-stratified analysis of the relationship between LPCAT3 and the risk of T2DM revealed a more pronounced negative correlation in males. Notably, in the BMI-stratified analysis, a negative correlation between LPCAT3 and T2DM risk was observed in the obese/overweight population but not in those with normal weight. Given that males have a higher BMI than females in our study, this finding aligns with the closer association between LPCAT3 and T2DM risk in males. However, this explanation does not hold when considering the results of an age-stratified analysis. Among participants, those aged <40 years had the highest BMI; however, no correlation was found between LPCAT3 and T2DM risk. In contrast, in the 40–60 age group, with a relatively lower BMI, a significant negative correlation between LPCAT3 and T2DM risk was observed, suggesting the presence of other underlying factors. The regression analysis stratified by age further complicates our understanding of the relationships between LPCAT3 and metabolic parameters (BMI, HDL and FBG). Correlations between LPCAT3 and

some variables were detected in the 40–60 age group and the ≥ 60 age group. Conversely, no correlations were found between LPCAT3 and BMI, HDL, or FBG in the < 40 age group. However, we must interpret the results of the age-stratified analysis with caution. To minimize selection bias, we consecutively recruited eligible newly diagnosed T2DM patients. Consequently, the 40–60 age group comprised the largest number of participants, which may suggest that this age range represents a high-incidence period for T2DM. In contrast, the relatively elderly population (≥60 years) had a smaller sample size, which may be attributed to strict screening criteria, such as excluding those who had taken medication within one month before enrollment. Additionally, elderly individuals frequently present with more complex comorbidities. These comorbidities have the potential to confound the observed correlation between LPCAT3 levels, metabolic parameters, and T2DM in this population. The age group under 40 had a relatively small participant count and presented with higher BMI values. These findings might suggest a comparatively lower prevalence of T2DM in the population under 40. Nevertheless, obesity remains a potential risk factor for T2DM. It is imperative to acknowledge that, due to the small sample size and the presence of complex health conditions, the contingency and complexity of the correlation results are likely to be elevated. To further address these issues, future studies ought to employ larger sample sizes, adopt well-conceived age-stratified research strategies, and integrate more all-encompassing observational indicators. These efforts will facilitate a more in-depth elucidation of the potential contributions of age and gender to the associations among LPCAT3, metabolic parameters, and the risk of T2DM. Comparisons of demographic and clinical parameters across different age groups are presented in S25 Table.

It is noteworthy that LPCAT3, beyond its involvement in metabolic pathways, plays a pivotal role in various physiological and pathological processes. Its complex dual effects on inflammatory responses and ferroptosis regulation are particularly striking. On one hand, it enhances membrane unsaturation, thereby alleviating endoplasmic reticulum stress induced by saturated free fatty acids (FFAs) and mitigating inflammatory responses and cellular damage from lipid peroxidation [36,37]. On the other hand, LPCAT3 increases PUFA-PC levels in the cell membrane, rendering the membrane more vulnerable to free radical attacks. This heightens the cell's susceptibility to ferroptosis and facilitates ferroptosis initiation [38,39]. These processes are intricately linked to diabetes [40] and lipid metabolism [41]. The question of whether alterations in LPCAT3 influence glucose and lipid metabolism via the inflammation and ferroptosis pathways remains an area of great interest and warrants further investigation.

It is remarkable that LPCAT3 is widely expressed in various tissues, including the liver and kidneys, where it plays a pivotal role in maintaining cell membrane integrity, supporting cellular function, and facilitating signal transduction. Its involvement in inflammatory responses and ferroptosis regulation further highlights its crucial role in modulating organ function, as supported by numerous studies [42,43]. In the liver, LPCAT3 exerts multifaceted effects. Some research underscores its role in promoting lipid synthesis, thereby influencing hepatic lipid metabolism [28,31], while other studies suggest it mitigates cellular damage caused by elevated lipid peroxidation [36,37]. A liver injury model involving liver-specific LPCAT3 knockout mice demonstrated reduced hepatic necrosis areas and attenuated elevations in serum transaminases (ALT and AST), indicating a potential protective role for LPCAT3 in liver health [44]. However, our current study found no significant correlations between serum LPCAT3 levels and hepatic function indicators such as AST and ALT, nor discernible differences in LPCAT3 levels between individuals with and without fatty liver disease, as detailed in S2 Table. Similarly, the role of LPCAT3 in the kidneys is complex and closely linked to ferroptosis. Overexpression of LPCAT3 in renal tubular epithelial cells leads to decreased cell viability and ferroptosis induction [45], and elevated LPCAT3 expression in the blood during acute kidney injury models underscores its potential involvement in kidney injury progression [46]. As a member of the lysophosphatidylcholine acyltransferase family, LPCAT3 may also influence renal hemodynamics by regulating the renal free arachidonic acid pool, thus affecting renal function [43]. Additionally, its upregulation in the livers of hyperuricemic mice suggests a role in hyperuricemia development and progression [25]. However, our study found no correlations between serum LPCAT3 levels and renal function markers such as creatinine (Cr) and uric acid (UA). In the context of atherosclerosis, LPCAT3 is essential for regulating phospholipid metabolism, cholesterol homeostasis,

cholesterol efflux, and inflammatory responses [47]. Hematopoietic cell-specific Lpcat3 knockout in mice exacerbates atherosclerosis [47], whereas macrophage-specific deletion has no substantial impact on atherogenesis [48]. In human atherosclerotic tissues, reduced LPCAT3 expression correlates with disease severity [49]. Nevertheless, our study found no differences in serum LPCAT3 levels between individuals with and without carotid atherosclerosis, as shown in S2 Table. Despite the lack of significant associations between serum LPCAT3 levels and parameters reflecting hepatic and renal function in our study, it is crucial to consider potential confounding factors, as previous research has emphasized the significance of various variables. In our study comparing the NGT and T2DM groups, significant differences were observed in several aspects, including ALT, Cr, fatty liver status, atherosclerosis status, and inflammatory indicators. These disparities raise the question of whether they could influence the observed differences in serum LPCAT3 levels between the two populations. For example, altered hepatic function, as reflected by changes in ALT levels, may be linked to the metabolic milieu in which LPCAT3 operates, potentially affecting its expression or activity. Similarly, variations in renal function, indicated by Cr levels, could have implications for the systemic regulation of LPCAT3. Moreover, the presence of fatty liver disease and atherosclerosis, which are associated with complex metabolic and inflammatory alterations, might indirectly impact LPCAT3 levels. Inflammatory indicators, often elevated in T2DM and related conditions, could also modulate LPCAT3 expression or function. Given these considerations, these factors warrant in-depth reflection in future research. Larger sample sizes, broader observation indicators, and dynamic multi-system observations may be essential for unraveling the complex relationships between LPCAT3 and these parameters. Such efforts could provide deeper insights into the biological roles and therapeutic potential of LPCAT3 in various pathological conditions.

## Conclusion and limitations

This study conducted a preliminary exploration of the alterations in serum LPCAT3 levels among patients with newly diagnosed T2DM and investigated its role in glucose and lipid metabolism. We observed reduced levels of serum LPCAT3 in these patients. However, the association between serum LPCAT3 and T2DM is not independent of adiposity; rather, it is likely confounded or mediated by obesity-related factors. Additionally, while there appears to be a tendency towards a negative correlation between serum LPCAT3 and FBG, BMI, and HDL levels, these relationships become nuanced when adjusting for confounding factors. Specifically, the interplay among glycemic indices, lipid profiles, and anthropometric measures may independently influence or modify the observed associations. Despite the study's findings, several limitations warrant consideration. Firstly, we must acknowledge the limited explanatory and predictive power of our models. The final stepwise linear regression model explained only 4.9% of the variance in serum LPCAT3 levels ($R^2 = 0.049$), suggesting that unmeasured factors are likely the main contributors of LPCAT3 variability. Similarly, the ROC curve analysis yielded an AUC of 0.580, indicating poor discriminatory performance. Furthermore, the instability of predictor relationships due to collinearity, as evidenced by sensitivity to variable inclusion, underscores the weak and unreliable nature of the identified associations. In addition, the substantial standard errors yielded by the PLS analysis provide further evidence of this uncertainty. Secondly, our study design targeted individuals with either NGT or diabetes to delineate distinct phenotypic differences between these two groups. This approach minimized potential confounding factors associated with intermediate metabolic states such as prediabetes, allowing us to confidently attribute observed differences in LPCAT3 levels to the unique metabolic characteristics of NGT and diabetic individuals. However, we acknowledge that excluding prediabetic subjects restricts our ability to comprehensively assess early metabolic changes preceding T2DM diagnosis, thereby hindering a thorough evaluation of LPCAT3's potential involvement in disease progression, particularly during the pivotal pre-diabetic stage. Thirdly, the cross-sectional nature of our study, which concentrates on serum LPCAT3 levels at a solitary time point, impedes the monitoring of longitudinal fluctuations in LPCAT3 expression over time or within tissues integral to T2DM pathophysiology, such as adipose tissue and the liver. This limitation curtails our capacity to fully elucidate LPCAT3's role in metabolic regulation. Fourthly, the recruitment of participants from four local medical institutions in the Jiangsu Taizhou region may introduce referral bias. It is plausible that the T2DM cohort includes a disproportionately

high number of severe cases, and individuals with heightened health consciousness or research participation enthusiasm might be overrepresented. Additionally, the continuous recruitment process and specific inclusion/exclusion criteria may have led to an overrepresentation of middle-aged individuals. Furthermore, the geographical location of the participating institutions and the socioeconomic environment of the local populace could have influenced the demographic and clinical profiles of the study volunteers. Fifthly, despite our efforts to match the NGT and T2DM groups for gender and age, inherent disparities in characteristics such as BMI, lipid profiles, blood pressure, organ and vascular health, dietary and exercise habits, and potential comorbidities persist. These confounding factors have the potential to skew our study outcomes. To mitigate these limitations, future investigations should adopt a longitudinal design with larger, more heterogeneous cohorts to capture the temporal and tissue-specific dynamics of LPCAT3 levels. Additionally, the utilization of multivariate modeling techniques, incorporating potential confounding variables (e.g., organ function, lifestyle factors) as covariates in regression analyses, will be pivotal in disentangling the independent impacts of glycemic indices, lipids, and anthropometric parameters on LPCAT3 levels. By employing these rigorous methodologies, researchers can ascertain whether LPCAT3 dysregulation precipitates metabolic dysfunction or is a consequence thereof, thus clarifying its role in metabolic health and disease. Such advancements will enhance our comprehension of LPCAT3's potential as a biomarker or therapeutic target in T2DM and associated metabolic disorders.

## Supporting information

**S1 Fig. Recruitment & study flow diagram.**
(TIF)

**S2 Fig. Visualization of predicted vs. observed LPCAT3 in PLS models with obesity-related anthropometric, lipid, and glucose indicators as predictors.**
(TIF)

**S1 Table. Comparison of serum LPCAT3 Levels across liver and carotid status groups.**
(DOCX)

**S2 Table. Evaluating the influence of confounding factors excluding obesity, lipid, and glucose parameters on a linear regression model for LPCAT3.**
(DOCX)

**S3 Table. Simple linear model considering only WHR as the independent variable.**
(DOCX)

**S4 Table. Incorporating WHR instead of BMI into the linear regression model.**
(DOCX)

**S5 Table. Incorporating both BMI and WHR as independent variables into the linear regression model.**
(DOCX)

**S6 Table. Incorporating WC instead of BMI into the linear regression model.**
(DOCX)

**S7 Table. Incorporating both BMI and WC as independent variables into the linear regression model.**
(DOCX)

**S8 Table. Incorporating all lipid-related indicators into the regression model.**
(DOCX)

**S9 Table. Incorporating 2hPG instead of FBG into the linear regression model.**
(DOCX)

**S10 Table. Incorporating HbA1c instead of FBG into the linear regression model.**
(DOCX)

**S11 Table. Incorporating HOMA-IR instead of FBG into the linear regression model.**
(DOCX)

**S12 Table. Incorporating both FBG and 2hPG as independent variables into the linear regression model.**
(DOCX)

**S13 Table. Incorporating both FBG and HbA1c as independent variables into the linear regression model.**
(DOCX)

**S14 Table. Incorporating all glucose-related indicators into the regression model.**
(DOCX)

**S15 Table. Selected cut-off points based on serum LPCAT3 levels for ROC analysis of T2DM prediction.**
(DOCX)

**S16 Table. Gender-stratified regression analysis of the association between LPCAT3 and metabolic parameters (BMI, HDL, FBG).**
(DOCX)

**S17 Table. Age-stratified regression analysis of the association between LPCAT3 and metabolic parameters (BMI, HDL, FBG).**
(DOCX)

**S18 Table. Logistic regression analysis of T2DM risk, with serum LPCAT3 as an independent predictor.**
(DOCX)

**S19 Table. Binary logistic regression assesses independent predictors of T2DM, including serum LPCAT3, demographic factors (sex, age, BMI), and their interactions.**
(DOCX)

**S20 Table. Independent predictors of T2DM risk by gender identified by binary logistic regression, with serum LPCAT3 levels as a key predictor.**
(DOCX)

**S21 Table. Independent predictors of T2DM risk by age group identified by binary logistic regression, with serum LPCAT3 levels as a key predictor.**
(DOCX)

**S22 Table. Independent predictors of T2DM risk identified by binary logistic regression, with serum LPCAT3 levels as a key predictor and stratified by BMI (cut-off at 24 kgm$^2$).**
(DOCX)

**S23 Table. Comparison of demographic and clinical parameters between male and female participants.**
(DOCX)

**S24 Table. BMI-stratified regression analysis of the association between LPCAT3 and metabolic parameters (BMI, HDL, FBG).**
(DOCX)

**S25 Table. Comparison of demographic and clinical characteristics among different age groups.**
(DOCX)

## Acknowledgments

We extend our sincerest gratitude to Zhou XL, Sun XP, and Ni M for their invaluable assistance in recruiting participants. We are deeply indebted to Zhou NX and her team for their substantial contributions to the data measurement process. During the revision phase of this manuscript, we are particularly grateful to Professor Cheng XB and his team, as well as many anonymous reviewers, for their insightful comments and constructive feedback. Their invaluable suggestions have greatly enhanced the quality and rigor of our work, and we are truly appreciative of their time and expertise.

## Author contributions

**Conceptualization:** Gaonian Zhao, Qian Li.

**Data curation:** Haifeng Zhu, Ziyi Zhong, Jing Jin, Wei Liu, Yuan Cao, Yawen Guo.

**Formal analysis:** Haifeng Zhu, Ziyi Zhong, Jing Jin, Wei Liu, Yuan Cao, Yawen Guo.

**Funding acquisition:** Gaonian Zhao.

**Investigation:** Haifeng Zhu, Ziyi Zhong, Jing Jin, Wei Liu, Yuan Cao, Yawen Guo.

**Methodology:** Haifeng Zhu, Ziyi Zhong, Jing Jin, Wei Liu, Yuan Cao, Yawen Guo.

**Project administration:** Haifeng Zhu, Gaonian Zhao, Qian Li.

**Resources:** Haifeng Zhu.

**Supervision:** Haifeng Zhu, Gaonian Zhao, Qian Li.

**Writing – original draft:** Haifeng Zhu, Ziyi Zhong, Jing Jin.

**Writing – review & editing:** Gaonian Zhao, Qian Li.

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
