## [Decision Letter · Decision Letter 0]

Dear Dr. Zhao,

Thank you for submitting your manuscript to PLOS ONE. After careful consideration, we feel that it has merit but does not fully meet PLOS ONE’s publication criteria as it currently stands. Therefore, we invite you to submit a revised version of the manuscript that addresses the points raised during the review process.

We look forward to receiving your revised manuscript.

Kind regards,

Hideto Sano

Academic Editor

PLOS ONE

Journal Requirements:

Reviewers' comments:

Reviewer's Responses to Questions

**Comments to the Author**

1. Is the manuscript technically sound, and do the data support the conclusions?

Reviewer #1: Partly

Reviewer #2: Yes

2. Has the statistical analysis been performed appropriately and rigorously?

Reviewer #1: Yes

Reviewer #2: Yes

3. Have the authors made all data underlying the findings in their manuscript fully available?

Reviewer #1: Yes

Reviewer #2: Yes

4. Is the manuscript presented in an intelligible fashion and written in standard English?

Reviewer #1: Yes

Reviewer #2: Yes

Reviewer #1: In the current study the authors explored the role of serum lysophosphatidylcholine acyltransferase 3 (LPCAT3) in glucose and lipid metabolism and investigated its association with type 2 diabetes mellitus. They underlined that lower serum LPCAT3 levels may be associated with an elevated risk of developing T2DM and that there is a tendency for serum LPCAT3 levels to negatively correlate with BMI, HDL, and fasting blood glucose.

Some suggestions:

1. In my opinion “glycolipid metabolism” is not suitable for a keyword.

2. Inclusion criteria, line 101 – 102: please check the statement “The study enrolled adults aged 18 to 80 years who had not used any medication in the preceding month”. Any medication for a 80 year patient?

3. Table 1: For a part of the biochemical parameters the reference values are not presented. Please complete.

4. Add please some details concerning the serum lysophosphatidylcholine acyltransferase 3 determination using ELISA method.

5. At discussion section are presented a lot of aspects taken from the literature but the results of the present study are insufficiently discussed.

6. In my opinion, incorporating prediabetic patients into the study enhanced the study's significance.

Reviewer #2: I would like to thank the editor for inviting me to review this article as this paper raises a very interesting question that will benefit clinical practice. However, I have to acknowledge some recommendations and comments throughout the manuscript.

The Title

Title need to be rewritten to be more indicative about the aim of the study.

The introduction:

-“The authors conduct very relevant research but fail to emphasize the relevance in their introduction and need to be more clarified in the introduction section. The exact cause of trying to assess if Serum Lysophosphatidylcholine Acyltransferase 3 Levels Correlated with Metabolic Variables and Predictive Value for Type 2 Diabetes Mellitus Risk is not clear enough, and need to be more clarified in the introduction section.

-Clarify primary and secondary outcome of your study at the end of introduction.

In method:

• Clarify the type of study design used, duration of the study and inclusion and exclusion criteria.

• If this study was approved by the ethics committee write the registration date and numbers.

• Also, please mention which international code of ethics are you adhering to?

• Add CONSORT flow chart for the study.

In Statistical Analysis:

-How did they control confounding? Did they assess for multicollinearity?

- How was sample size calculated?

In Results:

-Table 1 &2: Some abbreviations not elaborated under the table, Revise.

--Author should mention how data is presented, and the Statistical tests used for comparison and the level of significance below each table and figure.

-ROC-CURVE: authors should mention the cutting off points and 95 % confidence interval and the detailed table for area under the curve.

Discussion

- Lack of discussion regarding the effect of age, sex, and comorbidities can have within this study.

Limitations and conclusion:

- This study had limitations, primarily the issue of referral bias and possible confounding."

-Controlling for confounding variables is a critical step in conducting observational studies. The authors should use potential confounding variables as covariates in a regression model.

- the conclusion should be rewritten based on the above point.

**Do you want your identity to be public for this peer review?** For information about this choice, including consent withdrawal, please see our Privacy Policy

Reviewer #1: No

Reviewer #2: **Yes: ** Rehab H. Werida

---

## [Author Response · Author response to Decision Letter 1]

26 May 2025

Dear editors and reviewers,

We extend our heartfelt gratitude for the opportunity to revise our manuscript. We sincerely appreciate the insightful feedback and constructive criticism provided by the reviewers, whose expertise significantly enhanced the rigor and clarity of our work. Their suggestions—spanning experimental design, data analysis, and in-depth analytical scrutiny—were invaluable in refining our study.

Response to Reviewer A’s Comments:

1. We appreciate the reviewer's feedback on the keyword 'glycolipid'. To avoid confusion and more accurately reflect our study's focus, we have revised it to ‘lipid and glucose’.

2. We confirm that our study included adults aged 18 to 80 years who had not used any medication in the past 30 days to minimize potential confounding effects from medications on our metabolic outcome measures. We are fully aware that when recruiting potential participants, we encountered a relatively small number of eligible individuals aged 60 and above, despite this age group having the highest response rate. The stringent inclusion criteria, particularly the no-medication-use requirement, excluded many individuals who might otherwise have been interested in participating. This observation is valuable for future studies, as it highlights a potential limitation of our current approach. In future research, we will carefully consider the possibility of modifying the inclusion criteria. For instance, we might explore the feasibility of excluding only specific types of medications, such as antidiabetic drugs and other medications known to have a significant impact on the primary outcomes of interest. This adjustment could potentially increase the pool of eligible participants while still maintaining a reasonable level of control over confounding factors. Once again, we thank you for your constructive feedback, which has helped us to reflect on the strengths and limitations of our study design and has provided us with valuable insights for future research endeavors.

3. We have added the reference values for all biochemical parameters in Table 1, adhering to Chinese clinical standards.

4. We have included details on the serum LPCAT3 measurement method: Samples were diluted 1:4 with the kit buffer, and absorbance was measured at 450 nm using a standard curve ranging from 0 to 40 ng/ml.

5. Thank you very much for your constructive feedback. We've diligently revised the discussion section to emphasize our study's results and their comparison with existing literature.

6. We are extremely grateful for your thought-provoking suggestion to include prediabetic patients in our study. We recognize the importance of including prediabetic patients in our study to better understand LPCAT3's role in early glucose dysregulation. Although we identified a subgroup with impaired glucose tolerance (IGT) during OGTT screening. However, our study was originally conceived to focus on individuals with either normal glucose tolerance (NGT) or diabetes. This design was driven by our primary research questions, which necessitated clear-cut phenotypic differences to enable robust statistical comparisons. When planning the study, we gave priority to minimizing potential confounding factors linked to intermediate metabolic states, like prediabetes, to improve the interpretability of our results. Our aim was to sidestep selection bias and confidently attribute any observed differences in LPCAT3 levels to the distinct metabolic characteristics of the NGT and diabetic groups. While this strategy bolstered our primary analyses, we're fully aware that it restricted our capacity to thoroughly assess LPCAT3 across the entire spectrum of glucose metabolism. We whole-heartedly concur that future studies should incorporate prediabetic populations to fill this knowledge gap. Such research would offer invaluable insights into how LPCAT3 behaves during the transition from normal blood sugar levels to diabetes and its potential as a therapeutic target for early intervention. We sincerely thank you for pointing out this crucial avenue of research, and we'll make it a top priority in our future endeavors.

Response to Reviewer B’s Comments:

The title:

1. Thank you for your valuable suggestion. We have revised the title to more accurately reflect the study's cross-sectional design and objectives.

The introduction:

1. We have clarified the research gap regarding serum LPCAT3 and emphasized the importance of exploring its links with metabolic variables and T2DM risk.

2. We've clarified the primary (association between serum LPCAT3 levels and T2DM risk) and secondary outcomes (correlations with metabolic variables) at the end of the introduction as you suggested.

In method:

1. Thank you for emphasizing the need for clarity. This was a cross-sectional study conducted from July 1, 2024, to December 31, 2024. The inclusion and exclusion criteria are fully detailed in the Methods section, ensuring transparency and reproducibility.

2. We sincerely appreciate your attention to ethical compliance. Ethical approval was obtained before study initiation, and trial registration was completed. These details are now included in the Ethics Statement and Trial Registration sections.

3. This study adhered to the Declaration of Helsinki (2013 revision), ensuring participant rights and safety. A statement to this effect has been added to the Ethics Statement section.

4. We apologize for the oversight and thank you for the suggestion. We have included a CONSORT flow diagram (S1 Fig) in the supplementary material, detailing participant recruitment, screening, allocation, and follow-up. Due to the journal's possible figure-limit rules, we had to put it there. We hope for your understanding.

In statistical analysis:

1. Your suggestions regarding the control of collinearity and confounding factors are undoubtedly constructive. We consulted numerous senior researchers in the field, and all of them were astounded by your keen insight in this regard. They expressed concerns that our handling might not fully meet your high standards. Nonetheless, we are immensely grateful for the opportunity to have learned a great deal of professional knowledge from you through this process. We have made revisions to the Methods and Results sections based on your valuable feedback. We sincerely appreciate your guidance and the profound respect we hold for your expertise.

2. In response to your valuable suggestion, we have relocated the sample size estimation to the Methods section.

In results:

1. We've carefully explained all abbreviations that appear in the tables to ensure clarity.

2. Thank you for your guidance. We’ve now included detailed descriptions of data presentation methods, statistical tests used for comparisons, and the significance levels beneath each table and figure as suggested.

3. Thank you sincerely for your invaluable guidance on refining the ROC curve analysis. We’ve now incorporated the cutoff points, 95% confidence intervals, and a detailed table summarizing the area under the curve (AUC) results. Your expertise has been immensely educational, and we deeply appreciate the opportunity to improve our work under your supervision. If there are any remaining areas that require adjustment, please don’t hesitate to correct us—we hold your profound knowledge in the highest regard and are eager to ensure the manuscript meets your standards. Thank you again for your time and insight.

Discussion:

1. We sincerely appreciate your astute observation regarding the lack of discussion on the effects of age, sex, and comorbidities in our study. We fully recognize the importance of analyzing comorbidities. In response, we've added an analysis of liver and kidney function in the appropriate sections, as these are crucial comorbidity-related factors. During the study, we also collected liver and carotid artery ultrasound information. Although this wasn't part of our original research plan, many participants strongly requested these ultrasound examinations. So, we conducted them on a completely voluntary basis, informing the participants of their right to refuse to share the ultrasound results and obtaining their written consent. Initially, we decided not to include the ultrasound results in the manuscript because only a portion of the participants completed the examinations, and we didn't find any particularly significant findings at that time. It was a tough call to add these results during the review process, as we were concerned about potential negative consequences. However, we ultimately convinced all the authors to include them. This decision was not only a respectful and proactive response to your valuable suggestions but also driven by our realization that these results could be an important discovery, offering readers more insights and potentially benefiting future research. As for age and sex, we acknowledge that there has been relatively little research in this area in the past. To preserve the conciseness and flow of our manuscript, we opted not to delve into this aspect in detail. We sincerely appreciate your understanding of our decision, which was made with the aim of maintaining a clear and focused presentation of our findings. Once again, thank you for your insightful comments, which have undoubtedly helped us improve our study.

Limitations and conclusion:

1. We have revised our conclusion to acknowledge that while low serum LPCAT3 levels may be associated with an increased risk of T2DM, the underlying relationships are complex due to the interplay among glycemic indices, lipid profiles, and anthropometric measures. We acknowledge that our initial interpretation may have oversimplified these complexities, and we are grateful for your guidance in presenting them more accurately.

2. Regarding the study's limitations, we acknowledge the potential for referral bias due to our single-center, geographically limited sample, which may have led to an overrepresentation of certain patient groups. Additionally, we recognize the inadequacy in our control and analysis of confounding factors. For instance, we did not sufficiently account for variables such as diet, physical activity, body composition, blood lipid and pressure levels, organ function, and comorbidities. We sincerely regret this oversight and propose that future large-scale, diverse longitudinal studies utilize multivariate modeling to more comprehensively account for these factors. We sincerely apologize for any shortcomings in our study and are committed to learning from this experience. Your feedback will be invaluable in steering our future research efforts towards greater rigor and comprehensiveness.

We are immensely grateful for your invaluable insights. Given our current level of understanding, our revisions may still fall short in certain areas. We sincerely apologize for any oversights in our study and are committed to learning from this experience. We deeply appreciate the time and effort you have dedicated to reviewing our work.

Best regards,

Gaonian Zhao

---

## [Decision Letter · Decision Letter 1]

Dear Dr. Zhao,

Thank you for submitting your manuscript to PLOS ONE. After careful consideration, we feel that it has merit but does not fully meet PLOS ONE’s publication criteria as it currently stands. Therefore, we invite you to submit a revised version of the manuscript that addresses the points raised during the review process.

We look forward to receiving your revised manuscript.

Kind regards,

Hideto Sano

Academic Editor

PLOS ONE

Reviewers' comments:

Reviewer's Responses to Questions

**Comments to the Author**

Reviewer #1: All comments have been addressed

Reviewer #3: (No Response)

2. Is the manuscript technically sound, and do the data support the conclusions?

Reviewer #1: Yes

Reviewer #3: Partly

3. Has the statistical analysis been performed appropriately and rigorously?

Reviewer #1: Yes

Reviewer #3: Yes

4. Have the authors made all data underlying the findings in their manuscript fully available?

Reviewer #1: Yes

Reviewer #3: Yes

5. Is the manuscript presented in an intelligible fashion and written in standard English?

Reviewer #1: Yes

Reviewer #3: Yes

Reviewer #1: (No Response)

Reviewer #3: Thank you for the opportunity to review your manuscript, "Association of serum lysophosphatidylcholine acyltransferase 3 levels with metabolic variables and risk of type 2 diabetes mellitus: a cross-sectional study." The study addresses an interesting and important question regarding the role of serum LPCAT3 in metabolic health. The manuscript is well-structured, and I commend your efforts in responding to the previous round of reviews.

While the study is promising, there are several major areas that require significant revision before the manuscript can be considered for publication. My comments are intended to be constructive and to help you strengthen the manuscript.

Major Points:

1. Overarching Interpretation and the Confounding Effect of Obesity: This is the most critical issue with the manuscript. Your logistic regression analysis in Table 4 is the centerpiece of the study's primary outcome. While Models 1-4 show a significant association between the highest LPCAT3 tertile and lower T2DM risk, this association becomes non-significant (p=0.115) in Model 5 after adjusting for anthropometric indicators (BMI, WC, WHR).

o Conclusion: This result does not support the conclusion that "low serum LPCAT3 levels may be associated with an increased risk of developing T2DM." A more accurate conclusion is that the association between serum LPCAT3 and T2DM is not independent of adiposity and is likely confounded or mediated by it.

o Required Action: The abstract, discussion, and conclusion must be substantially rewritten to reflect this finding. The narrative should shift from "LPCAT3 as a risk factor" to "LPCAT3 as a potential marker of a metabolic state related to obesity, which itself is a risk factor for T2DM." This is a more nuanced and scientifically accurate interpretation of your data.

2. Weak Explanatory and Predictive Power of Models:

o In your stepwise linear regression (Table 3), the final model has an R-squared value of 0.049. This indicates that your independent variables (BMI, HDL, FBG) explain only 4.9% of the variance in serum LPCAT3 levels. This is extremely low and suggests that other, unmeasured factors are the primary determinants of LPCAT3 levels.

o Similarly, the ROC curve analysis (Fig 2) yields an AUC of 0.580. This represents very poor discriminatory power, barely better than a coin toss (AUC=0.5).

o Required Action: While you report these numbers, their implications must be stated more explicitly in the Discussion and Limitations sections. You must acknowledge that, based on your data, the clinical utility of serum LPCAT3 as a predictive or explanatory biomarker appears to be minimal.

3. Omission of Age and Sex-Specific Analyses: In your response to Reviewer B, you state that you opted not to delve into age and sex analyses "to preserve the conciseness and flow of our manuscript." For a study on metabolic disease, this is a major omission, not a matter of conciseness. Age and sex are fundamental modifiers of metabolism, adiposity, and T2DM risk.

o Required Action: You must perform and present analyses stratified by sex. For example, do the correlations between LPCAT3 and BMI/HDL/FBG hold for both men and women? You should also test for an interaction between LPCAT3 and sex/age on T2DM risk. These analyses are standard practice and essential for a rigorous study.

Section-Specific Comments:

• Abstract:

o The conclusion needs to be revised to reflect the non-significance of the association after adjusting for obesity. Example: "Conclusion: In this cohort, the association between lower serum LPCAT3 levels and T2DM was not independent of obesity. While LPCAT3 correlates with measures of adiposity and glycemic control, its predictive power for T2DM is limited."

o Introduction

o "With the progress in metabolomics and lipidomics research, the critical role of lipid metabolism dysregulation in the onset and progression of T2DM has gained increasing recognition " add this reference to this sentence DOI: 10.18502/aacc.v10is2.17213

• Methods:

o The exclusion of individuals with pre-diabetes is a major design choice that should be justified more robustly in the main manuscript's Discussion/Limitations, not just in the reviewer response. It fundamentally limits your ability to discuss LPCAT3's role in the progression from normoglycemia to T2DM.

• Results:

o Your reporting of the complex collinearity analyses (S3-S14 Tables) is commendable. However, the key takeaway from this—that the relationships are unstable and dependent on which correlated variable is included in the model—should be summarized in the main text of the results to give the reader a clearer sense of the data's complexity.

o The reporting of the PLS analysis is honest about the large standard errors. This reinforces the conclusion that the observed correlations are weak and unreliable.

• Discussion:

o The discussion should be restructured. A significant portion should be dedicated to exploring why the association between LPCAT3 and T2DM is attenuated by obesity. You could hypothesize that LPCAT3 is more directly involved in adipocyte biology or lipid storage regulation, and its link to T2DM is therefore indirect.

o You rightly contrast your findings with tissue-specific studies. This is a strong part of the discussion. You can strengthen it further by discussing the potential sources of serum LPCAT3 (e.g., liver, adipose tissue) and how systemic levels might not reflect the biology of a single tissue.

o Remove speculative phrasing that is not supported by your weak data. Focus on what your data clearly show and what they do not.

I believe that addressing these points will significantly improve the quality and impact of your manuscript. The research topic is of high interest, and a more cautious and deeply analyzed interpretation of your findings will make a valuable contribution to the field.

**Do you want your identity to be public for this peer review?** For information about this choice, including consent withdrawal, please see our Privacy Policy

Reviewer #1: No

Reviewer #3: No

---

## [Author Response · Author response to Decision Letter 2]

1 Jul 2025

Dear Dr. Hideto Sano and Reviewers,

We are truly grateful for the invaluable opportunity to revise our manuscript. We sincerely acknowledge that in the previous round, we handled some issues rather hastily. The professional and insightful comments from the reviewers have been extremely beneficial, allowing us to gain a deeper understanding of the shortcomings in our study. Due to our limited expertise, there may still be some inadequacies, and we deeply appreciate the time and effort you have invested in reviewing our work. We apologize for any inconvenience caused.

First and foremost, we must openly admit that there was improper data handling in the previous version of the manuscript. During the logarithmic conversion of non-normally distributed data, we adopted different bases, using 10 in some instances and the natural number e in others. This lack of consistency not only produced erratic results but also posed a significant hurdle for readers in discerning the relationships among the data. In this round of revisions, we have standardized all logarithmic conversions to use the natural number e as the base. We have clearly articulated this in the Methods section of the article and provided supplementary explanations in the footnotes of specific tables to aid readers in comprehending the results. Notably, certain coefficients in S2-14 have been revised. We have clearly marked these modifications and uploaded the modification records in a separate folder. We are acutely aware that this may cause substantial inconvenience during your review process. We sincerely apologize for our previous improper handling.

The following modifications have been made in response to the concerns raised by the reviewers.

Major Points:

1. We sincerely appreciate the reviewer's sharp insight into the confounding effect of obesity. We fully recognize that our prior way of expressing the related results was not proper. We have carefully reworked the relevant parts to make them more scientifically and rigorously presented.

2. We acknowledge the weak explanatory and predictive power of our models. We have explicitly stated these implications in the Discussion and Limitations sections. We acknowledge that, based on our data, the clinical utility of serum LPCAT3 as a predictive or explanatory biomarker appears to be minimal.

3. We recognize that the omission of age- and sex-specific analyses was a significant oversight. Age and sex are fundamental modifiers of metabolism, adiposity, and T2DM risk. We have now performed and presented analyses stratified by sex. We have tested whether the correlations between LPCAT3 and BMI/HDL/FBG hold for both men and women, and also tested for an interaction between LPCAT3 and sex/age on T2DM risk. These sections are located in the last paragraph of the Results section. We deeply regret our reckless oversight of this issue during the previous review round and are immensely grateful for your kindness in granting us another opportunity to revise. Given the differing views on how to address the issue, we consulted with several experts for their insights and, after much deliberation, finally reached a consensus. We are genuinely concerned about whether our handling is appropriate. If this has caused you any inconvenience during the review, we sincerely apologize.

Section-Specific Responses

Abstract

As mentioned above, we have revised the conclusion in the abstract to reflect the non-significance of the association after adjusting for obesity.

Introduction

We have added the reference (DOI: 10.18502/aacc.v10is2.17213) to the sentence "With the progress in metabolomics and lipidomics research, the critical role of lipid metabolism dysregulation in the onset and progression of T2DM has gained increasing recognition."

Methods

In the Limitations section, we've dissected the reasons for not considering pre-diabetes in the experimental design. We fully acknowledge that this design decision significantly impedes our holistic grasp of the glycemic spectrum.

Results

We have summarized the key takeaway from the complex collinearity analyses (S3-S14 Tables) in the main text of the results. This gives the reader a clearer sense of the data's complexity, specifically that the relationships are unstable and dependent on which correlated variable is included in the model.

We have briefly summarized the results of the PLS analysis and acknowledge that the observed correlations are weak and unreliable.

Discussion

During the revision process, we actively sought counsel from several experts. They unanimously conveyed their deep-seated respect for your professionalism and, notably, sang high praises for your modification suggestions regarding the Discussion section. We are profoundly grateful for your invaluable insights. Your statement “Focus on what your data clearly show and what they do not” was truly an eye-opener for us. Although we still harbor some uncertainty about the appropriateness of our revisions, we can distinctly perceive a substantial enhancement in the Discussion section, all thanks to your invaluable guidance. Your assistance is, without a doubt, an irreplaceable treasure for our research endeavor.

We explored the reasons why obesity attenuates the association between LPCAT3 and T2DM. Building on this, we hypothesized that the connection between LPCAT3 and T2DM is indirect, as it is influenced by factors like obesity, and we subsequently provided a brief discussion on this hypothesis.

We conducted a brief analysis on the potential differences in the origin and function of serum LPCAT3 compared to tissue-specific LPCAT3.

We have removed speculative phrasing that is not supported by our weak data.

The final manuscript was reviewed by two independent experts. They made some minor adjustments to the sentence expressions, aiming to minimize potential confusion for readers and enhance the fluency of the article. We acknowledge that these revisions may necessitate additional scrutiny during your re-examination. For any inconvenience this may cause, we extend our most sincere apologies, and express profound gratitude for your unwavering commitment to upholding the intellectual integrity of this scholarly endeavor.

The supervision of the revision process for this manuscript was undertaken by Professor Xingbo Cheng. He is a respected elder in the local community and has twice served as the chairman of the Jiangsu Diabetes Society. He specifically instructed us to convey to you his utmost respect for your outstanding professional knowledge and inclusive and magnanimous qualities.

During the revision process, we observed a marked improvement in the manuscript's quality. This valuable experience has not only deepened our knowledge but also laid a solid groundwork for our future research. Indeed, this submission has proven to be a highly constructive endeavor. We extend our heartfelt gratitude to the editors and all reviewers for their unwavering commitment to enhancing this manuscript. Your meticulous attention to detail and profoundly insightful suggestions have been of immense value to our work. However, acknowledging the limits of our expertise, we hesitate to assert that we've fully addressed all your concerns in the revision. We sincerely apologize for any inconvenience this may have caused and would greatly appreciate your continued guidance and support.

Sincerely,

Xingbo Cheng, Gaonian Zhao, Haifeng Zhu, et al. (on behalf of all the authors)

2025-07-01

---

## [Decision Letter · Decision Letter 2]

Association of serum lysophosphatidylcholine acyltransferase 3 levels with metabolic variables and risk of type 2 diabetes mellitus: a cross-sectional study

PONE-D-25-16090R2

Dear Dr. Zhao,

We’re pleased to inform you that your manuscript has been judged scientifically suitable for publication and will be formally accepted for publication once it meets all outstanding technical requirements.

Kind regards,

Hideto Sano

Academic Editor

PLOS ONE

Reviewers' comments:

Reviewer's Responses to Questions

**Comments to the Author**

Reviewer #1: All comments have been addressed

Reviewer #2: All comments have been addressed

Reviewer #3: All comments have been addressed

2. Is the manuscript technically sound, and do the data support the conclusions?

Reviewer #1: Yes

Reviewer #2: Yes

Reviewer #3: Yes

3. Has the statistical analysis been performed appropriately and rigorously?

Reviewer #1: Yes

Reviewer #2: Yes

Reviewer #3: Yes

4. Have the authors made all data underlying the findings in their manuscript fully available?

Reviewer #1: Yes

Reviewer #2: Yes

Reviewer #3: Yes

5. Is the manuscript presented in an intelligible fashion and written in standard English?

Reviewer #1: Yes

Reviewer #2: Yes

Reviewer #3: Yes

Reviewer #1: (No Response)

Reviewer #2: The authors have addressed all my concerns. I consider the manuscript is now acceptable for publication.

Reviewer #3: I read the new version of manuscript and authors could answer all of my comments. So, I do not have additional comments.

**Do you want your identity to be public for this peer review?** For information about this choice, including consent withdrawal, please see our Privacy Policy

Reviewer #1: No

Reviewer #2: **Yes: ** Rehab H. Werida, Faculty of Pharmacy, Damanhour University, Egypt.

Reviewer #3: **Yes: ** Mehran Rahimlou

---

## [Editor Report · Acceptance letter]

PONE-D-25-16090R2

PLOS ONE

Dear Dr. Zhao,

I'm pleased to inform you that your manuscript has been deemed suitable for publication in PLOS ONE. Congratulations! Your manuscript is now being handed over to our production team.

Kind regards,

on behalf of

Dr. Hideto Sano

Academic Editor

PLOS ONE